# DELTA-LoRA: FINE-TUNING HIGH-RANK PARAMETERS WITH THE DELTA OF LOW-RANK MATRICES

## ABSTRACT

In this paper, we present **Delta-LoRA**, which is a novel parameter-efficient approach to fine-tune large language models (LLMs). In contrast to LoRA and other low-rank adaptation methods such as AdaLoRA, Delta-LoRA not only updates the low-rank matrices $A$ and $B$, but also propagate the learning to the pre-trained weights $W$ via updates utilizing the delta of the product of two low-rank matrices $(A^{(t+1)}B^{(t+1)} - A^{(t)}B^{(t)})$. Such a strategy effectively addresses the limitation that the incremental update of low-rank matrices is inadequate for learning representations capable for downstream tasks. Moreover, as the update of $W$ does not need to compute the gradients of $W$ and store their momentums, Delta-LoRA shares comparable memory requirements and computational costs with LoRA. Extensive experiments show that Delta-LoRA significantly outperforms existing low-rank adaptation methods. We further support these results with comprehensive analyses that underscore the effectiveness of Delta-LoRA.

## 1 INTRODUCTION

Large Language Models (LLMs) recently have attracted considerable attention due to their remarkable performance across a broad spectrum of downstream tasks. Diverging from conventional Transformers characterized by a scale of millions of parameters, modern LLMs typically scale up to billions of parameters, endowing them with notable advantages such as emergent capabilities and robust generalization as detailed in (Bubeck et al., 2023). Fine-tuning such highly capable LLMs on downstream tasks (Raffel et al., 2020; Devlin et al., 2019; Radford et al., 2019; He et al., 2021; Liu et al., 2019; Brown et al., 2020) has consequently become a mainstream paradigm to reduce the training time required for individual tasks, yet with superior performance compared with other methods (Lester et al., 2021; Li & Liang, 2021; Houlsby et al., 2019).

However, fine-tuning a LLM with all the learnable parameters (Full Fine-tuning) requires multiple GPUs with high memory demand (Dettmers et al., 2023; Hu et al., 2022), which is unattainable for many companies and research institutions. Full fine-tuning poses exceptional challenges to researchers: with massive parameter size, LLMs already demand more storage space than regular models; Further training exaggerates the GPU memory requirement because common optimizers such as AdamW (Loshchilov & Hutter, 2019) often maintain several copies of the model parameters, which is 2-3 times of memory overhead.

To this end, a series of methods have been proposed (Valipour et al., 2023; Zhang et al., 2022; Li & Liang, 2021; Liu et al., 2022a; Lv et al., 2023; Dettmers et al., 2023; Liu et al., 2022b; Zaken et al., 2021; Pfeiffer et al., 2021; Guo et al., 2021; Zhou et al., 2023; Zhang et al., 2023; Houlsby et al., 2019; Wang et al., 2022) to reduce memory overhead at the training stage. Some even accelerate the fine-tuning process with only less than $1\%$ trainable parameters. Among these methods, LoRA (Hu et al., 2022) is the most attractive for its stable performance on broad downstream tasks (Ding et al., 2023), no observed overfitting, as well as no extra memory and computation cost at inference.

While LoRA and its successors (Zhang et al., 2022; Valipour et al., 2023) have indeed exhibited superior performance in comparison to alternative approaches within the realm of Parameter Efficient Fine-Tuning (PEFT), a substantial **performance gap** persists when compared to the full fine-tuning, as highlighted in most scenarios (Ding et al., 2023). This discrepancy is attributed to the inherent limitation of updating only a fraction of the model's parameters, rendering it inadequate to fit the intricacies presented in the training data.

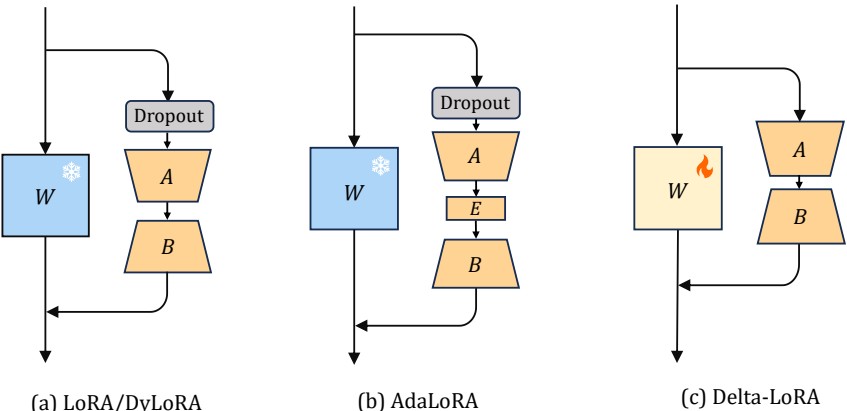

(a) LoRA/DyLoRA       (b) AdaLoRA       (c) Delta-LoRA

Figure 1: An overview of the proposed **Delta-LoRA** structure, compared to **LoRA**, **DyLoRA** and **AdaLoRA**. Note that **DyLoRA** and **LoRA** basically share the same architecture. $W$ is the pre-trained weight which is frozen (signified by blue) when performing efficient-parameter fine-tuning in (a) and (b). Orange trapezoids $A$, $B$ and $E$ denote the trainable parameters. In our proposed Delta-LoRA, the light orange rectangle means that pre-trained weights can be updated via the delta. *Note that our proposed Delta-LoRA removes the Dropout layer to ensure reasonable delta for pre-trained matrix.*

To bridge this gap, a reasonable strategy is to introduce more parameters into the optimization process. In this paper, we introduce Delta-LoRA as shown in Fig. 1, a novel PEFT approach that simultaneously updates the pre-trained matrix and two low-rank matrices while maintaining the same memory consumption as the original LoRA. Specifically, the pre-trained matrix $W$ is updated with the delta of the product of two low-rank matrices in two consecutive iterations ($\triangle AB = A^{(t+1)}B^{(t+1)} - A^{(t)}B^{(t)}$), while two low-rank matrices are updated by the AdamW optimizer automatically. This is based on the mathematical property that $\frac{\partial \mathcal{L}}{\partial W} = \frac{\partial \mathcal{L}}{\partial AB}$ and $\triangle AB$ is a surrogate to direct the update of $W$ (see Sec. 4 for details). Since we neither store the gradient of $W$ nor use the optimizer to update the pre-trained matrix, the proposed method thus does not yield any extra memory overhead. This strategic integration effectively mitigates the sub-optimal representation learning stemming from only updating the two low-rank matrices. Moreover, our approach aligns the update direction of the pre-trained weights with that of the incremental update matrix. Furthermore, we discard the Dropout layer in low-rank branches to obtain a more reasonable delta for $W$, in order to ensure $\frac{\partial \mathcal{L}}{\partial W} = \frac{\partial \mathcal{L}}{\partial AB}$. The advantages of our proposed method are conspicuous: including the pre-trained weights in the optimization process engenders a broader integration of parameters, thereby enhancing the potential for learning intricate representations.

The main contributions of this paper can be summarized as:

- We introduce Delta-LoRA, a novel PEFT method that simultaneously updates the full weight matrix and two low-rank matrices. Delta-LoRA leverages the delta of the product of $A$ and $B$ to update the pre-trained weights and thus prevent storing the first and the second-order momentums in the optimizer.

- We analyze the gradient flow of Delta-LoRA and show that the Dropout layer in the low-rank branch makes $\frac{\partial \mathcal{L}}{\partial W} \neq \frac{\partial \mathcal{L}}{\partial AB}$. Thus, we remove the Dropout layer in our proposed Delta-LoRA to get reasonable delta for $W$.

- We conduct comprehensive experiments to show that Delta-LoRA has consistent gains on a broad range of NLP tasks. Additionally, we provide thorough explanations to analyze its superiority and the value contributed by each component.

## 2   PRELIMINARIES

**Transformer-based Models.** Transformer (Vaswani et al., 2017) adopts the self-attention mechanism instead of recurrence and convolutions, achieving new state-of-the-art in machine translation. Dosovitskiy et al. (2021) later proposed the Vision-Transformer (ViT) architecture which exhibits

versatility across various computer vision tasks. Nowadays, the Transformer-based models have become the most popular choice in both NLP and Computer Vision (Li et al., 2021; Carion et al., 2020; Zheng et al., 2021). Transformer typically consists of $L$ stacked blocks, each containing a multi-head attention (MHA) module and a feed-forward network (FFN) module. For an input sequence $\boldsymbol{X} \in \mathbb{R}^{n \times d}$, the MHA module yields the output MHA($\boldsymbol{X}$), given by:

$$\text{head}_i = \text{softmax}\left(\frac{\boldsymbol{X}\boldsymbol{W}_{Q_i}(\boldsymbol{X}\boldsymbol{W}_{K_i})^\top}{\sqrt{d_k}}\right)\boldsymbol{X}\boldsymbol{W}_{V_i}$$

$$\text{MHA}(\boldsymbol{X}) = \text{concat}(\text{head}_1, ..., \text{head}_k)\boldsymbol{W}_o, \tag{1}$$

where $d_k$ is the scaling factor and set to $d_k = d/k$. $\boldsymbol{W}_{K_i}$, $\boldsymbol{W}_{Q_i}$, $\boldsymbol{W}_{V_i}$ and $\boldsymbol{W}_o$ are weight matrices for computation of key, query, value and the output of MHA, respectively. Besides the MHA module, the FFN is also vital in the Transformer-based model. It stacks two fully connected (FC) layers with an activation function in between. FFN is defined as:

$$\text{FFN}(\boldsymbol{x}) = \boldsymbol{W}_{f_2}\text{ReLU}(\boldsymbol{W}_{f_1}\boldsymbol{x} + \boldsymbol{b}_1) + \boldsymbol{b}_2, \tag{2}$$

where $\boldsymbol{x} \in \mathbb{R}^d$, $\boldsymbol{W}_{f_1}$ and $\boldsymbol{W}_{f_2}$ are two fully connected layers in FFN, $\boldsymbol{b}_1$ and $\boldsymbol{b}_2$ are bias terms.

**Low Rank Adaptation**. Given a pre-trained matrix $\boldsymbol{W} \in \mathbb{R}^{c \times d}$, LoRA (Hu et al., 2022) learns an incremental update $\triangle\boldsymbol{W}$ and decomposes $\triangle\boldsymbol{W}$ into a matrix multiplication between two low-rank matrices $\boldsymbol{A}$ and $\boldsymbol{B}$, where $\boldsymbol{A} \in \mathbb{R}^{c \times r}$ and $\boldsymbol{B} \in \mathbb{R}^{r \times d}$, and $\triangle\boldsymbol{W} = \boldsymbol{A}\boldsymbol{B}$. Here, the rank $r \ll min(d, c)$. For an input $\boldsymbol{x}$ and hidden state $\boldsymbol{h}$, LoRA has the following forward process:

$$\boldsymbol{h} = \boldsymbol{W}^*\boldsymbol{x} = \boldsymbol{W}\boldsymbol{x} + \triangle\boldsymbol{W}\boldsymbol{x} = \boldsymbol{W}\boldsymbol{x} + \frac{\alpha}{r}\boldsymbol{A}\boldsymbol{B}\boldsymbol{x} \tag{3}$$

At the beginning of the training stage, $\boldsymbol{A}$ is randomly initialized via Kaiming initialization (He et al., 2015) and $\boldsymbol{B}$ is initialized to zero matrix to make sure that the incremental update $\boldsymbol{A}\boldsymbol{B} = \boldsymbol{0}$ at initialization. Besides, LoRA uses hyper-parameters $\alpha$ and $r$ to scale $\boldsymbol{A}\boldsymbol{B}\boldsymbol{x}$.

## 3 RELATED WORKS

With the ever-growing parameter scale in current Transformer-based models, fine-tuning such a large language model (LLM) requires considerable number of GPUs equipped with high memory capacity. This is mainly due to the fact that common optimizers such as AdamW (Loshchilov & Hutter, 2019) requires maintaining three times of extra parameter size (gradients, first-order and second-order momentums). To bridge this gap, a series of Parameter-Efficient Fine-Tuning (PEFT) methods have been proposed (Hu et al., 2022; Liu et al., 2022b; Shin et al., 2020; Houlsby et al., 2019). The Adapter (Houlsby et al., 2019) introduces lightweight trainable parameters between pre-trained layers while keeping the pre-trained weights fixed. Prompt-Tuning (Lester et al., 2021) aims to optimize the prompt to achieve comparable performance with fine-tuning for specific task, while Prefix-Tuning optimizes for trainable prefixes and prepends these trainable parameters to each hidden state (Li & Liang, 2021). Despite the notable performance achievements, these methods inevitably introduce extra overhead at the inference stage.

Hu et al. (2022) proposed LoRA to utilize the multiplication of two low-rank matrices to model the incremental update of a full-rank matrix. LoRA merges the incremental updates to pre-trained weights after training, thereby avoiding any extra computation overhead during inference. Furthermore, it stands out as one of the most effective PEFT techniques according to Ding et al. (2023)'s evaluation. Subsequent to its inception, a series of enhanced methods building upon LoRA was proposed. Notably, G-LoRA (Chavan et al., 2023) leverages a generalized prompt module to fine-tune pre-trained weights resulting in better representations for computer vision tasks. DyLoRA (Valipour et al., 2023) aims to adjust the rank of two lightweight matrices after the training stage. Differing from the conventional approach of maintaining a static rank during training, DyLoRA introduces rank variations to its blocks. AdaLoRA (Zhang et al., 2022) emphasizes the disparate importance attributed to distinct weight parameters. This technique intelligently allocates the parameter budget across weight matrices based on their respective importance scores. Additionally, Q-LoRA (Dettmers et al., 2023) was proposed to further reduce the average memory footprint by quantizing the pre-trained model with 4-bit NormalFloat. This quantization approach not only preserves the model's efficacy but also effectively alleviates the resource-intensive nature of LLM training and addresses a pertinent concern.

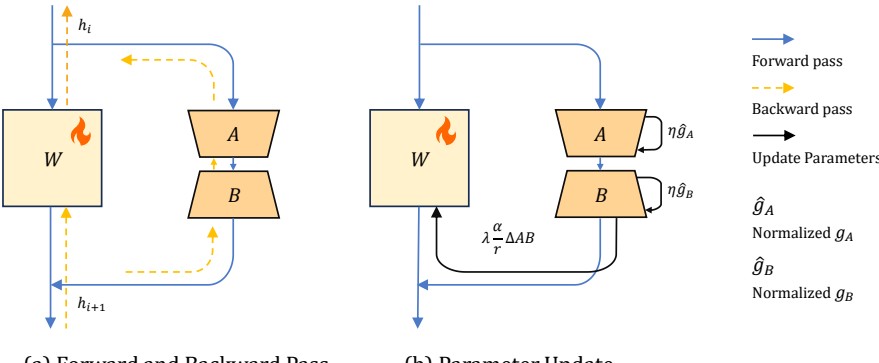

Figure 2: The framework of our proposed Delta-LoRA. The blue arrows represent forward pass while yellow dashed arrows denote backward propagation. The black solid arrows in (b) represent the process of updating the low-rank adaptation matrices $A$ and $B$ with normalized gradients $\widehat{g}_A$ and $\widehat{g}_B$ multiplied by the learning rate $\eta$, as well as updating the pre-trained weights $W$ with the delta matrix $\triangle AB$ multiplied by the update ratio $\lambda$.

## 4 METHODOLOGY

This section introduces the novel fine-tuning approach termed as Delta-LoRA. Delta-LoRA encompasses two pivotal designs as shown in Figure 1 and Figure 2: (i) It simultaneously updates the full weight matrix ($W$) alongside the two low-rank adaptation matrices ($A$ and $B$), utilizing the delta ($A^{(t+1)}B^{(t+1)} - A^{(t)}B^{(t)}$) resulting from incremental updates to refine the pre-trained weights ($W$); (ii) The Dropout layer as originally integrated within the conventional LoRA module, is excluded in Delta-LoRA. This omission stems from the realization that its presence violates the required assumption $\frac{\partial \mathcal{L}}{\partial W} = \frac{\partial \mathcal{L}}{\partial AB}$.

### 4.1 UPDATE THE DELTA OF LOW-RANK MATRICES ON PRE-TRAINED WEIGHTS

For an input $x$ and its corresponding hidden state $h$, LoRA optimizes two low-rank matrices $A$ and $B$ to learn an incremental update $AB$ for the pre-trained and fixed weight matrix $W$. Different from previous methods, we argue that $W$ also needs to be updated. In this way, we can introduce more learnable parameters to the optimization process for higher learning capability. However, acquiring the normalized gradients (i.e. the gradients after normalization in optimizer) to fine-tune the weight matrix $W$ is non-trivial, since the optimizer such as AdamW must maintain at least three extra copies of the parameters (i.e. gradients as well as the first-order and the second-order moments of gradients) in GPU memory. Intriguingly, we note that the gradients of the loss $\mathcal{L}$ with respect to matrices $AB$ and $W$ are precisely identical, under the presumption that the LoRA module exclusively retains matrices $A$ and $B$, while disregarding the Dropout layer. This correspondence can be formally represented as:

$$
\begin{aligned}
g_W &= \frac{\partial \mathcal{L}}{\partial h_{i+1}} \cdot \frac{\partial h_{i+1}}{\partial W}^{\top} = \frac{\partial \mathcal{L}}{\partial h_{i+1}} \cdot h_i^{\top}, \\
g_{AB} &= \frac{\partial \mathcal{L}}{\partial h_{i+1}} \cdot \frac{\partial h_{i+1}}{\partial AB}^{\top} = \frac{\partial \mathcal{L}}{\partial h_{i+1}} \cdot h_i^{\top}, \\
&\Longrightarrow g_W = g_{AB},
\end{aligned}
\tag{4}
$$

where $h_{i+1} = Wh_i + ABh_i$, $h_i$ and $h_{i+1}$ are the outputs of the $i$-th layer and the $i+1$-th layer respectively. $AB$ is the matrix product of the adaptation matrices $A$ and $B$, $\mathcal{L}$ is the loss function, while $g_{W+AB}$, $g_W$ and $g_{AB}$ denote the gradients of $\frac{\partial \mathcal{L}}{\partial (W+AB)}$, $\frac{\partial \mathcal{L}}{\partial W}$, and $\frac{\partial \mathcal{L}}{\partial AB}$ respectively.

Equation 4 inspires us to use $g_{AB}$ to assimilate $g_W$ when learning the parameter updates for weight matrix $W$. Unfortunately, we are only able to obtain the gradients $g_A$ and $g_B$ rather than $g_W$ during the back-propagation process. Furthermore, the computation of the gradients for $AB$ is as expensive

as for the matrix $W$, since both matrices share the same dimensions of $d \times k$, consequently entailing an equivalent GPU memory overhead.

Considering a typical optimization process, the model updates its parameters by applying the gradient descent: $W^{(t+1)} = W^{(t)} - \eta g_W$, with the parameter update denoted as $\triangle W = -\eta g_W$, using the learning rate $\eta$. Similarly, we regard $-\triangle AB$ as the gradients for $AB$ and utilize this matrix as a substitute for $g_W$ according to Equation 4. Here, we can compute $\triangle AB$ as:

$$\triangle AB = A^{(t+1)} B^{(t+1)} - A^{(t)} B^{(t)} = \eta A^{(t)} g_B + \eta g_A B^{(t)} - \eta^2 g_A g_B, \qquad (5)$$

where $A^{(t)}$, $B^{(t)}$ and $W^{(t)}$ are the weights of $A$, $B$ and $W$ at the $t$-th step respectively, $A^{(t+1)} = A^{(t)} - \eta g_A$, $B^{(t+1)} = B^{(t)} - \eta g_B$ and $\eta$ is the learning rate. To be precise, $-\triangle AB$ does not equate directly to $g_{AB}$ and $g_W$ as elaborated in Appendix A.4.2. Nonetheless, $\triangle AB$ has the capability to symbolize the genuine directions of update for the matrix $AB$. Based on this assumption, it is reasonable to employ $-\triangle AB$ as the gradient for directing the update of $W$.

Therefore, during the training phase we introduce the matrix $\triangle AB$ to update the pre-trained weights $W$ in the following manner:

$$W^{(t+1)} = W^{(t)} + \lambda \cdot \frac{\alpha}{r} \cdot \triangle AB, \text{ where } \triangle AB = A^{(t+1)} B^{(t+1)} - A^{(t)} B^{(t)}, \qquad (6)$$

where $\lambda$ represents the hyper-parameter to trade off the update ratio of $AB$ and the pre-trained weights $W$. The parameter updates for $W$ commence after $K$ training iterations. The procedural details of the algorithm are illustrated in Algorithm 1.

**Discussion.** The Delta-LoRA has some important modifications compared to LoRA. Here, we discuss and compare the difference:

| LoRA | Delta-LoRA |
|---|---|
| (1) $A^{(t+1)} \leftarrow \frac{\partial \mathcal{L}(x; W, A^{(t)}, B^{(t)})}{\partial A^{(t)}}$ | (1) $A^{(t+1)} \leftarrow \frac{\partial \mathcal{L}(x; W^{(t)}, A^{(t)}, B^{(t)})}{\partial A^{(t)}}$ |
| (2) $B^{(t+1)} \leftarrow \frac{\partial \mathcal{L}(x; W, A^{(t)}, B^{(t)})}{\partial B^{(t)}}$ | (2) $B^{(t+1)} \leftarrow \frac{\partial \mathcal{L}(x; W^{(t)}, A^{(t)}, B^{(t)})}{\partial B^{(t)}}$ |
| | (3) $W^{(t+1)} \leftarrow A^{(t+1)} B^{(t+1)} - A^{(t)} B^{(t)}$ |

It is obvious that LoRA only updates $A$ and $B$, and keeps $W$ frozen, while Delta-LoRA updates $A$ and $B$ by the optimizer and $W$ with the delta of the product of $A$ and $B$.

## 4.2 THE STRUCTURE OF OUR DELTA-LORA

Both LoRA and its successor AdaLoRA put a Dropout layer before two low-rank matrices $A$ and $B$. However, this arrangement results in a disparity between the gradient matrices $g_W$ and $g_{AB}$ (or the matrix $g_{AEB}$ in the context of AdaLoRA). The derivation of this disparity can be shown as:

$$g_W = \frac{\partial \mathcal{L}}{\partial h_{i+1}} \cdot h_i^\top \neq g_{AB} = \frac{\partial \mathcal{L}}{\partial h_{i+1}} \cdot \text{Drop}(h_i)^\top, \qquad (7)$$

where $\text{Drop}(\cdot)$ denotes the Dropout layer which leads to $g_W \neq g_{AB}$. A reasonable choice is to remove the Dropout layer in the low-rank module and activate the Dropout layer between pre-trained layers if overfitting problem occurs. This modification also brings additional benefits: (1) it can alleviate under-fitting to some extent, thereby enhancing the learned representations of the networks. The rationale behind this improvement lies in the fact that LoRA and its successors formulate low-rank updates for pre-trained weights, involving less than $1\%$ of the complete parameters. However, relying solely on such a small fraction of parameters may not bestow an adequate representation capacity in most cases; (2) This alteration also yields memory-saving benefits. By negating the requirement to store intermediate features, the model curtails the memory consumption. Consequently, there is a reduction in activation memory employed during the back-propagation process.

## 5 EXPERIMENTS

We evaluate our proposed model fine-tuning method Delta-LoRA with RoBERTa (Liu et al., 2019), GPT-2 (Radford et al., 2019) and BART (Lewis et al., 2019) on a broad set of datasets. Specifically,

we train (1) RoBERTa on GLUE benchmark which consists of 8 NLP understanding tasks; (2) GPT-2 on E2E Challenge and WebNLG Challenge 2017 following the setting of Hu et al. (2022); and (3) BART on XSum dataset by using the setting provided by Zhang et al. (2022). See Appendix A.7 for more training details on the datasets. The setups and detailed introductions of baseline methods are shown in Appendix A.1. We use *PyTorch* to implement our experiments and download the pre-trained weights as well as configuration files from *HuggingFace* Wolf et al. (2019).

Table 1: The evaluation results of our proposed Delta-LoRA and other existing methods on E2E NLG Challenge dataset. † indicates fine-tuning all layers except embedding layer. ‡ indicates only fine-tuning weights for query and value. ¶ means we choose different settings with AdaLoRA: we only tune $W_Q$ and $W_V$ instead of all layers. The best results of Fine-Tuning methods are underlined. The best results of PEFT methods are **boldfaced**.

| Method | Trainable Parameters | Extra Updatable Parameters | BLEU | NIST | METEOR | ROUGE-L | CIDEr |
|---|---|---|---|---|---|---|---|
| Full Fine-Tuning | 354.92M | ✗ | 69.58 | 8.75 | 46.34 | 71.66 | 2.47 |
| Fine-Tuning† | 305.84M | ✗ | 69.37 | 8.76 | 46.05 | 71.97 | 2.44 |
| Fine-Tuning‡ | 48M | ✗ | 69.77 | 8.84 | 46.29 | 71.96 | 2.49 |
| LoRA (repr.) | 0.375M | ✗ | 69.60 | 8.78 | 45.61 | 71.12 | 2.45 |
| LoRA | 0.35M | ✗ | 70.4 | 8.85 | 46.8 | 71.8 | 2.53 |
| DyLoRA | 0.375M | ✗ | 67.89 | 8.50 | 44.07 | 70.52 | 2.26 |
| AdaLoRA¶ | 0.375M | ✗ | 68.16 | 8.58 | 44.10 | 70.66 | 2.35 |
| Delta-LoRA (Ours) | 0.375M | ✓ 48M | **70.84** | **8.91** | **46.47** | **72.24** | **2.53** |

Table 2: The evaluation results of our proposed Delta-LoRA and other existing methods on WebNLG Challenge 2017 dataset. † indicates fine-tuning all layers except embedding layer. ‡ indicates only fine-tuning weights for query and value. ¶ means we choose different settings with AdaLoRA: we only tune $W_Q$ and $W_V$ instead of all layers. The best results of Fine-Tuning methods are underlined. The best results of PEFT methods are **boldfaced**.

| Method | Trainable Parameters | Extra Updatable Parameters | BLEU↑ | | | METEOR↑ | | | TER↓ | | |
|---|---|---|---|---|---|---|---|---|---|---|---|
| | | | S | U | A | S | U | A | S | U | A |
| Full Fine-Tuning | 354.92M | ✗ | 61.38 | 45.11 | 54.48 | 0.44 | 0.38 | 0.41 | 0.36 | 0.53 | 0.44 |
| Fine-Tuning† | 305.84M | ✗ | 63.53 | 46.66 | 55.92 | 0.45 | 0.39 | 0.42 | 0.34 | 0.49 | 0.41 |
| Fine-Tuning‡ | 48M | ✗ | 64.55 | 48.06 | 57.08 | 0.46 | 0.39 | 0.43 | 0.33 | 0.47 | 0.40 |
| LoRA (repr.) | 0.375M | ✗ | 62.08 | 46.60 | 55.05 | 0.44 | 0.38 | 0.41 | 0.35 | 0.49 | 0.42 |
| LoRA | 0.375M | ✗ | 62.1 | 46.7 | 55.3 | 0.44 | 0.38 | 0.41 | 0.33 | 0.46 | 0.39 |
| DyLoRA | 0.375M | ✗ | 58.39 | 46.02 | 52.77 | 0.42 | 0.37 | 0.40 | 0.38 | 0.49 | 0.43 |
| AdaLoRA¶ | 0.375M | ✗ | 56.39 | 44.14 | 50.82 | 0.41 | 0.37 | 0.39 | 0.40 | 0.49 | 0.44 |
| Delta-LoRA (Ours) | 0.375M | ✓48M | **62.87** | **47.68** | **55.96** | **0.45** | **0.39** | **0.42** | **0.34** | **0.48** | **0.40** |

## 5.1 NATURAL LANGUAGE GENERATION

**Models and Datasets.** We use GPT2-Medium to verify the effectiveness of our Delta-LoRA on two datasets for data-to-text tasks, including the E2E NLG Challenge (Puzikov & Gurevych, 2018) and WebNLG Challenge 2017 (Gardent et al., 2017). The E2E NLG Challenge dataset comprises 42,000 samples for training, 4,600 for validation, and 4,600 for testing purposes. In contrast, the WebNLG Challenge 2017 consists of 21,855 training samples across nine categories, expanding to a total of 14 categories in the test set. For the text summarization task, we employed BART-Large (Lewis et al., 2019) to evaluate the efficacy of our method using the XSum dataset (Narayan et al., 2018). The XSum dataset is composed of 204,045 training samples, 11,332 validation samples, and 11,332 test samples. We also use LLaMA-7B(Touvron et al., 2023), a popular pre-trained large language model with 7 Billion parameters,to fine-tune on Alpaca dataset (Taori et al., 2023).
**Implementation Details.** In order to compare with LoRA and its successors fairly, we adopt the model setups from LoRA to implement our Delta-LoRA and three PEFT methods. We only learn

Table 3: The evaluation results of Delta-LoRA with LLaMA-7B on the Instruction-Tuning dataset provided by Stanford Alpaca(Taori et al., 2023). We use GPT-4 to choose from *a. LoRA*, *b. Delta-LoRA* or *c. Both LoRA and Delta-LoRA* to decide the text from which method is better.

| Both | LoRA | Delta-LoRA | Total |
|------|------|------------|-------|
| 886  | 10   | 104        | 1,000 |

the low-rank incremental update for $W_Q$ and $W_V$ in MHA module. For data-to-text datasets, we use the same training configurations as adopted by LoRA, including the number of training epochs, batch size and etc. We use update ratio $\lambda = 2$ and set start steps $K = 500$ for Delta-LoRA. More details about Delta-LoRA are listed in the Appendix A.7. For the text-summarization task, we use the implementation of AdaLoRA and adopt the same training configurations. We set the update ratio $\lambda = 0.5$ and the start steps $K = 1000$ for Delta-LoRA.

**Experimental Results.** Table 1 shows the results for E2E Challenge dataset on 5 evaluation metrics, demonstrating that our method achieves state-of-the-art performance over 3 baselines and a set of fine-tuning methods. For the BLEU and ROUGE-L metrics, our method obtains 1.24 and 1.13 performance gains compared with LoRA, with 0.13, 0.86 and 0.08 improvement on NIST, METEOR and CIDEr respectively. Table 2 demonstrates that Delta-LoRA outperforms baselines on BLEU score for WebNLG Challenge 2017 dataset, with 0.79, 1.08 and 0.91 improvement on Seen, Unseen and All test data, respectively. Additionally, for the METEOR and TER evaluation metrics, Delta-LoRA also achieves state-of-the-art performance, with 0.01 and 0.02 improvement over LoRA on all data. For the text-summarization task, the test results are shown in Table 4, which demonstrates that our method achieves state-of-the-art results across 3 parameter-efficient methods on 4 evaluation metrics. To fairly evaluate our method, we utilized LLaMA-7B and compared it with LoRA. We employed GPT-4 to generate 1,000 questions and presented these questions to the parameter-efficient fine-tuned LLaMA-7B. Subsequently, we leveraged GPT-4 to compare the texts generated by LoRA-tuned and Delta-LoRA-tuned LLMs. Additional details can be found in Appendix A.2. According to the findings in Table 3, Delta-LoRA establishes state-of-the-art performance in the evaluation of Language Models (LLMs). GPT-4 identified 104 samples generated by Delta-LoRA as superior to LoRA, while only 10 samples generated by LoRA exhibited higher quality than Delta-LoRA. This underscores Delta-LoRA's effectiveness even when utilized within models containing billions of parameters.

Table 4: The evaluation results of our proposed Delta-LoRA and other existing methods on XSum dataset. † indicates fine-tuning all layers except the embedding layer. ‡ indicates only fine-tuning weights for query and value. ¶ means we choose different settings with AdaLoRA: we only tune $W_Q$ and $W_V$ instead of all layers. The best results of Fine-Tuning methods are underlined. The best results of PEFT methods are **boldfaced**.

| Method | Trainable Parameters | Extra Updatable Parameters | Rouge-1 | Rouge-2 | Rouge-L | Rouge-Sum |
|--------|---------------------|----------------------------|---------|---------|---------|-----------|
| Full Fine-Tuning | 387.5M | ✗ | 45.36 | 22.16 | 37.23 | 37.24 |
| Fine-Tuning† | 338.4M | ✗ | 45.04 | 22.05 | 36.92 | 36.94 |
| Fine-Tuning‡ | 72M | ✗ | 44.95 | 21.43 | 36.35 | 36.37 |
| LoRA | 0.56M | ✗ | 43.27 | 20.13 | 35.12 | 35.12 |
| DyLoRA | 0.56M | ✗ | 41.84 | 18.76 | 33.56 | 33.57 |
| AdaLoRA¶ | 0.56M | ✗ | 42.91 | 19.76 | 34.71 | 34.72 |
| Delta-LoRA (Ours) | 0.56M | ✓72M | **43.49** | **20.23** | **35.26** | **35.26** |

## 5.2 NATURAL LANGUAGE UNDERSTANDING

**Models and Datasets.** We use RoBERTa-baseLiu et al. (2019) to evaluate the performance of our proposed method, prior works and three fine-tuning methods. We choose the GLUE benchmark consisting of 8 datasets (Wang et al., 2019), including classification tasks, similarity and paraphrase tasks and natural language inference tasks.

**Implementation Details.** We use RoBERTa-base with 118M parameters to conduct our experiments and to compare our method with the baselines. We mostly adopt the same training configurations of

LoRA, more details can get from Appendix A.7. We set the rank to 8 and the target rank to 6 for AdaLoRA and choose the rest of hyper-parameters according to the characteristics of different tasks. For Delta-LoRA, we set the update ratio $\lambda$ to 0.5 and choose different start steps $K$ according to warmup steps used in individual tasks.

Table 5: The evaluation results of our proposed Delta-LoRA and other existing methods on GLUE benchmark. We report the overall (matched and mismatched) accuracy for MNLI, Matthew's correlation for CoLA, Pearson correlation for STS-B, and accuracy for other tasks. † indicates fine-tuning all layers except the embedding layer. ‡ indicates only fine-tuning weights for query and value. ¶ means we choose different settings with AdaLoRA: we only tune $W_Q$ and $W_V$ instead of all layers. The best results of Fine-Tuning methods are underlined. The best results of PEFT methods are **boldfaced**.

| Method | Trainable Parameters | Extra Updatable Parameters | MNLI | SST-2 | MRPC | CoLA | QNLI | QQP | RTE | STS-B | AVG |
|---|---|---|---|---|---|---|---|---|---|---|---|
| Full Fine-Tuning | 118.87M | ✗ | 87.51 | 94.26 | 88.23 | 64.57 | 92.73 | 91.96 | 84.11 | 90.56 | 86.74 |
| Fine-Tuning† | 82.05M | ✗ | 87.58 | 94.03 | 89.95 | 62.99 | 92.73 | 91.90 | 86.64 | 90.22 | 87.01 |
| Fine-Tuning‡ | 13.5M | ✗ | 87.48 | 95.06 | 89.21 | 61.07 | 92.76 | 91.19 | 84.83 | 89.85 | 86.43 |
| LoRA | 0.28M | ✗ | 87.40 | 94.61 | 89.95 | 63.17 | 93.02 | 90.67 | 86.64 | 91.54 | 87.12 |
| DyLoRA | 0.28M | ✗ | 86.33 | 94.26 | 89.46 | 61.12 | 92.22 | 90.17 | 84.47 | 91.06 | 86.14 |
| AdaLoRA¶ | 0.28M | ✗ | 87.34 | 94.49 | 90.19 | 61.64 | 93.08 | 90.14 | 85.19 | 91.16 | 86.65 |
| Delta-LoRA (Ours) | 0.28M | ✓ 13.5M | **87.50** | **95.06** | **90.19** | **63.82** | **93.09** | **90.87** | **87.00** | **91.57** | **87.38** |

**Experimental Results.** We compare our method with prior PEFT works. According to Table 5, our method outperforms existing methods on all 8 tasks in GLUE benchmark. Among these tasks, our method demonstrates significant improvement on SST-2, CoLA and RTE. This is mainly due to the fact that these datasets contain less training data, which hinders the model's capacity to effectively acquire a robust representation when using prior fine-tuning methods. Delta-LoRA also achieves decent performance on the rest of the datasets, including MNLI, MRPC, QNLI as well STS-B, which proves that our method is stable and reliable across different settings.

## 5.3 COMPREHENSIVE UNDERSTANDING OF DELTA-LORA

Table 6: The ablation study of our proposed Delta-LoRA on E2E Challenge dataset demonstrates the importance of each component. The best results are **boldfaced**.

| Method | Trainable Parameters | Extra Updatable Parameters | BLEU | NIST | METEOR | ROUGE-L | CIDEr |
|---|---|---|---|---|---|---|---|
| LoRA (repr.) | 0.375M | ✗ | 69.60 | 8.78 | 45.61 | 71.12 | 2.45 |
| Delta-LoRA + LoRA Module | 0.375M | ✓ 48M | 70.29 | 8.88 | 46.38 | 71.88 | 2.51 |
| Delta-LoRA | 0.375M | ✓ 48M | **70.84** | **8.91** | **46.47** | **72.24** | **2.53** |

Table 7: The ablation study of our proposed Delta-LoRA to eliminate the impact of hyper-parameter $\lambda$ on E2E Challenge dataset. The best results are **boldfaced**.

| Method | Learning Rate | $\lambda$ | BLEU | NIST | METEOR | ROUGE-L | CIDEr |
|---|---|---|---|---|---|---|---|
| LoRA (repr.) | 2e-4 | - | 69.60 | 8.78 | 45.61 | 71.12 | 2.45 |
| LoRA (repr.) | 6e-4 | - | 69.63 | 8.79 | 45.70 | 71.55 | 2.39 |
| Delta-LoRA | 2e-4 | 2 | **70.84** | **8.91** | **46.47** | **72.24** | **2.53** |

**The Extra Updatable Parameters.** We introduce the concept of extra updatable parameters to point out the superiority of Delta-LoRA. For most PEFT methods, they can only adjust the low-rank adapters, such as AdapterHoulsby et al. (2019) and LoRAHu et al. (2022). Thus, they don't have any extra parameters to update, which means their extra updatable parameters are 0. However, our Delta-LoRA can achieve the purpose of updating the $W$ matrix without increasing the GPU memory consumption, which means its extra updatable parameters are the parameter number of $W$.

**Ablation study.** To better understand the contribution of our modified LoRA module (i.e. Delta-LoRA module) and the effectiveness of our update algorithm, we conduct studies on E2E Challenge

dataset with GPT2-medium. As shown in Table 6, only updating the pre-trained matrices with delta of low-rank update can indeed achieve performance improvement, while further discarding the dropout in Delta-LoRA module obtains the best performance. This observation confirms the indispensable role played by each component within our proposed methodology. We have devised an experiment to further differentiate whether the performance enhancement stems from the inherent characteristics of our method rather than solely from the substantial update magnitude. According to our algorithm, we update the parameters of both pre-trained and low-rank matrices, which can arose the doubt of whether the improvement is caused by updating larger $\triangle AB$ on the weights instead of introducing more parameters into the optimization process. To answer this question, we design an experiment with results shown in Table 7 to prove the effectiveness of our method. We scale the learning rate of LoRA from 2e-4 to 6e-4 making sure that $W + AB$ can be updated with $3 \times \triangle AB$, which is equivalent to Delta-LoRA when $\lambda$ is set to 2. We find that even by updating with $3 \times \triangle AB$ on $AB$, the performance is still not comparable with Delta-LoRA. This experiment further proves that introducing more parameters into the optimization process can force to learn better representation.

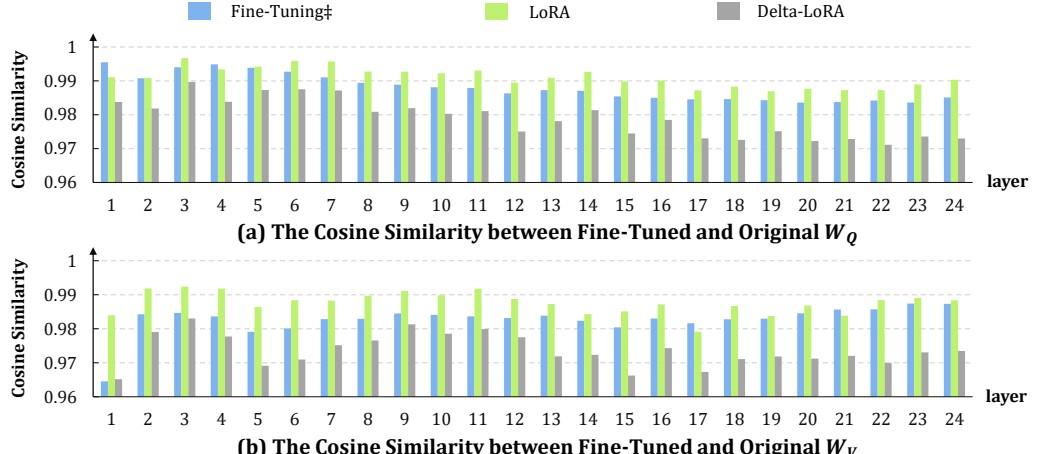

Figure 3: The comparison of Fine-Tuning‡, LoRA as well as Delta-LoRA for the cosine similarity between the fine-tuned parameters and the original pre-trained parameters in each transformer block. *Higher value means higher similarity.*

**The cosine similarity between fine-tuned and the pre-trained parameters to measure learning effects.** We conduct a comparative analysis of three methods including Fine-Tuning‡, LoRA and Delta-LoRA, in order to elucidate the reasons behind Delta-LoRA's superior performance. We use the last checkpoint trained on E2E Challenge dataset to give understanding. As depicted in Figure 3, it is evident that LoRA exhibits the highest similarity across the majority of transformer blocks. This observation suggests that LoRA primarily modifies the matrix $W^* = W + AB$ within a limited range. Nevertheless, Delta-LoRA showcases the lowest cosine similarity, underscoring that our approach induces the most significant modifications to the final matrix $W^*$. Due to this property, our approach can effectively stimulate the model to acquire better representations, leading to state-of-the-art performance across all four PEFT methods. This observation further aligns with the evaluation results in Table 1: Delta-LoRA achieves the best performance among the three methods, whereas LoRA is slightly worse than Fine-Tuning‡.

# 6 CONCLUSION

In this paper, we have introduced Delta-LoRA, a novel method to simultaneously update the full weight matrix and two low-rank matrices. Delta-LoRA leverages the delta $(A^{(t+1)}B^{(t+1)} - A^{(t)}B^{(t)})$ to update the pre-trained weights $(W)$. In this way, we introduce more learnable parameters into the optimization process such that the model can learn a better representation with comparable memory cost as LoRA. Meanwhile, we identify the Dropout layer in the low-rank branch to be unnecessary according to the gradient flow. We also provide thorough analysis of our method to understand its effectiveness and robustness. Extensive experiments on a broad range of NLP tasks are conducted to empirically verify the effectiveness of our Delta-LoRA.

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

# A APPENDIX

## A.1 BASELINES

We compare our proposed method Delta-LoRA with Fine-Tuning and prior works of LoRA, AdaLoRA, and DyLoRA. For PEFT methods, we only train the incremental updates for $W_V$ and $W_Q$, following the setup as used in LoRA's paper. For Fine-Tuning methods, we use two extra training paradigms: (1) freeze the embedding and train all the other parameters as Fine-Tuning †; (2) train $W_V$ and $W_Q$ only as Fine-Tuning‡.

**Fine-Tuning.** In the past few years, fine-tuning has become the mainstream paradigm for both NLP and CV tasks. However, fine-tuning full parameters is subject to potential drawbacks including overfitting and training instability (Huang et al., 2022). Therefore, freezing a subset of network layers and fine-tuning the rest has become a popular choice (Tan et al., 2018). In our experiments, we compare with full fine-tuning, fine-tuning with embedding layers frozen (Fine-tuning †) and fine-tuning query and value matrices only (Fine-tuning ‡).

**LoRA** (Hu et al., 2022) uses multiplication of two low-rank matrices to learn the incremental updates with reduced GPU memory cost. We follow their setups to reproduce experimental results for fair comparison.

**DyLoRA** (Valipour et al., 2023) randomly chooses a rank $r$ for LoRA modules during learning.

**AdaLoRA** (Zhang et al., 2022) focuses on the challenge of determining the optimal rank for incremental updates. It employs an adaptive approach to singular value pruning, tailoring the rank selection to the magnitude of each singular value. Consequently, distinct ranks are employed for different layers.

## A.2 THE COMPARISON BETWEEN LoRA AND DELTA-LoRA WITH LLaMA-7B

### A.2.1 TRAINING AND INFERENCE ARGUMENTS USED IN OUR METHOD AND BASELINE

We choose LLaMA-7B to evaluate our method and LoRA. Here, we set the learning rate $\gamma$ =1e-4, batch size to 128, $r = 8$, $\alpha$ =16, and training epochs to 3 for both two methods. Following the LoRA's paper, we only tune $W_Q$ and $W_V$. For Delta-LoRA, we choose start steps $K = 100$ and $\lambda = 0.25$. When inference, we set the no_repeat_ngram_size = 10, temperature = 0 and beam size = 4 to get a certain answer.

### A.2.2 THE EVALUATION FOR OUR METHOD AND BASELINES

Current LLMs obtain the training data from the Internet, which may unintentionally cause data leakage. Therefore, using the mainstream NLP datasets to evaluate the effectiveness of Large Language Model is not reasonable and wisdom. Inspired by evaluation approach proposed by Liu et al. (2023), we decided to use GPT-4 to judge the text generated by which method is accurate. First, we ask GPT-4 to generate 1,000 different questions. Second, we use the LLaMA-7B trained by two methods to generate the texts. Finally, we ask GPT-4 to give decision to tell us which text is accurate. It can choose from three options: *a. Choice 1* (LoRA generates accurate text), *b. Choice 2* (Delta-LoRA generates accurate text) and *c. Both Choice 1 and 2* (Both LoRA and Delta-LoRA generate accurate texts). The prompt we used for evaluation:
*Help me to determine which text is accurate for the given instruction and question. The answer can be chosen from a. Choice 1 is accurate, b. Choice 2 is accurate or c. both Choice 1 and 2 are accurate. Give me a certain answer and this is a choice question. Please don't give reasons and the answer must be shorter than 20 words.*
*Question: ""*
*(Choice 1): ""*
*(Choice 2): ""*

## A.3 ALGORITHM OF DELTA-LoRA

Our Delta-LoRA can be found in Algorithm 1. Compared to LoRA, we added a step to update the pre-trained $W$ without any extra GPU memory consumption.

---

**Algorithm 1:** Delta-LoRA

---

**Input:** Learning rate $\eta$; weight decay $\beta$; total training iterations $T$; low rank $r$; scale factor $\alpha$; start steps $K$; update ratio $\lambda$.

$A$ is initialized by Kaiming Initialization, $B = 0$ and $W$ is initialized with pre-trained weights.

**for** $t = 0, ..., T - 1$ **do**

Sample a mini-batch and compute gradients for $\{A, B\}$ in each Delta-LoRA module.

Update the first and second moments maintained by the optimizer with the computed gradients, and get the normalized gradients $\widehat{g}_A$ and $\widehat{g}_B$.

$A^{(t+1)} \leftarrow A^{(t)} - \eta \widehat{g}_A - \eta \beta A^{(t)}$

$B^{(t+1)} \leftarrow B^{(t)} - \eta \widehat{g}_B - \eta \beta B^{(t)}$

**if** $t > K$ **do**

$W^{(t+1)} \leftarrow W^{(t)} + \lambda \cdot \frac{\alpha}{r} \cdot (A^{(t+1)} B^{(t+1)} - A^{(t)} B^{(t)})$

**end if**

**end for**

**Output:** the fine-tuned parameters $\{W^{(T)}, A^{(T)}, B^{(T)}\}$

---

## A.4 A Further Understanding of Delta-LoRA

### A.4.1 The Differences between LoRA and Delta-LoRA

There are some fundamental differences between LoRA and Delta-LoRA.

• Given $W + AB$, $W$ is fixed in LoRA, but $W$ will be updated in our Delta-LoRA. This is **the largest difference between LoRA and Delta-LoRA**. This modification can yield more training differences between LoRA and Delta-LoRA in the next few training steps.

• $\text{Rank}(\Delta W_{Delta-LoRA}) = \text{Rank}(W^{(T)} - W^{(0)} + AB) > \text{Rank}(\Delta W_{LoRA}) = \text{Rank}(AB)$. The rank of the learned incremental weight matrix in our Delta-LoRA is larger than that in the original LoRA.

• The gradient flow is different between LoRA and Delta-LoRA. Suppose that we have $W \in \mathbb{R}^{m \times n}$, $A \in \mathbb{R}^{m \times r}$ and $B \in \mathbb{R}^{r \times n}$, where $r \leq \min(m, n)$. For LoRA, it keeps $W$ frozen, so that $W^* = W^{(0)} + \frac{\alpha}{r} \cdot A^{(t)} B^{(t)}$. For Delta-LoRA, it updates all matrices, and has $W^* = W^{(t)} + \frac{\alpha}{r} \cdot A^{(t)} B^{(t)}$.

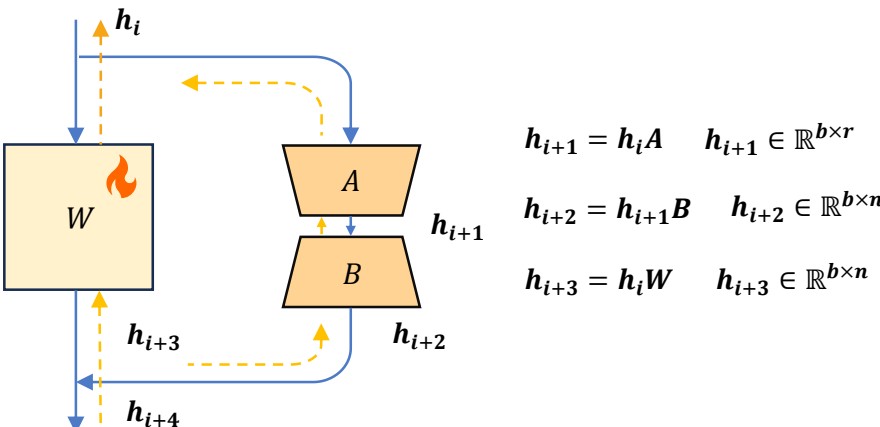

$$h_{i+1} = h_i A \qquad h_{i+1} \in \mathbb{R}^{b \times r}$$

$$h_{i+2} = h_{i+1} B \qquad h_{i+2} \in \mathbb{R}^{b \times n}$$

$$h_{i+3} = h_i W \qquad h_{i+3} \in \mathbb{R}^{b \times n}$$

Figure 4: The backward propagation of Delta-LoRA.

So, we have the following equation according to Figure 4:

$$\frac{\partial \mathcal{L}}{\partial W^{(t)}} = \left(\frac{\partial h_{i+3}}{\partial W^{(t)}}\right)^\top \frac{\partial \mathcal{L}}{\partial h_{i+4}} = h_i^\top \frac{\partial \mathcal{L}}{\partial h_{i+4}}$$

$$\frac{\partial \mathcal{L}}{\partial A^{(t)}} = \left(\frac{\partial h_{i+1}}{\partial A^{(t)}}\right)^\top \frac{\partial \mathcal{L}}{\partial h_{i+4}} \cdot \frac{\partial h_{i+4}}{\partial h_{i+2}} \cdot \frac{\partial h_{i+2}}{\partial h_{i+1}} = h_i^\top \frac{\partial \mathcal{L}}{\partial h_{i+4}} \frac{\partial h_{i+2}}{\partial h_{i+1}} = h_i^\top \frac{\partial \mathcal{L}}{\partial h_{i+4}} B^{(t)\top} = \frac{\partial \mathcal{L}}{\partial W^{(t)}} \cdot B^{(t)\top}$$

$$\frac{\partial \mathcal{L}}{\partial B^{(t)}} = \left(\frac{\partial \boldsymbol{h}_{i+1}}{\partial \boldsymbol{B}^{(t)}}\right)^{\top} \cdot \frac{\partial \mathcal{L}}{\partial \boldsymbol{h}_{i+4}} \cdot \frac{\partial \boldsymbol{h}_{i+4}}{\partial \boldsymbol{h}_{i+2}} = \boldsymbol{h}_{i+2}^{\top} \cdot \frac{\partial \mathcal{L}}{\partial \boldsymbol{h}_{i+4}} = (\boldsymbol{A}^{(t)})^{\top} \cdot \boldsymbol{h}_i^{\top} \cdot \frac{\partial \mathcal{L}}{\partial \boldsymbol{h}_{i+4}} = (\boldsymbol{A}^{(t)})^{\top} \cdot \frac{\partial \mathcal{L}}{\partial \boldsymbol{W}^{(t)}}$$

Here, we provide the back-propagation process of LoRA:

$$\frac{\partial \mathcal{L}}{\partial A^{(t+1)}} = \frac{\partial \mathcal{L}}{\partial W^{(t+1)}} \cdot (B^{(t+1)})^{\top} = \frac{\partial \mathcal{L}}{\partial (W^{(0)} + (A^{(t)} + \Delta A^{(t)})(B^{(t)} + \Delta B^{(t)}))} \cdot (B^{(t)} + \Delta B^{(t)})^{\top}$$

$$\frac{\partial \mathcal{L}}{\partial B^{(t+1)}} = (A^{(t+1)})^{\top} \cdot \frac{\partial \mathcal{L}}{\partial W^{(t+1)}} = (A^{(t)} + \Delta A^{(t)})^{\top} \cdot \frac{\partial \mathcal{L}}{\partial (W^{(0)} + (A^{(t)} + \Delta A^{(t)})(B^{(t)} + \Delta B^{(t)}))}$$

This is the back-propagation process of Delta-LoRA:

$$\frac{\partial \mathcal{L}}{\partial A^{(t+1)}} = \frac{\partial \mathcal{L}}{\partial W^{(t+1)}} \cdot (B^{(t+1)})^{\top} = \frac{\partial \mathcal{L}}{\partial ((W^{(t)} + \lambda \Delta A^{(t)} B^{(t)}) + (A^{(t)} + \Delta A^{(t)})(B^{(t)} + \Delta B^{(t)}))} \cdot (B^{(t)} + \Delta B^{(t)})^{\top}$$

$$\frac{\partial \mathcal{L}}{\partial B^{(t+1)}} = (A^{(t+1)})^{\top} \cdot \frac{\partial \mathcal{L}}{\partial W^{(t+1)}} = (A^{(t)} + \Delta A^{(t)})^{\top} \cdot \frac{\partial \mathcal{L}}{\partial ((W^{(t)} + \lambda \Delta A^{(t)} B^{(t)}) + (A^{(t)} + \Delta A^{(t)})(B^{(t)} + \Delta B^{(t)}))}$$

### A.4.2 THE EXPANSION OF $\triangle AB$

In the real training process, we need to consider a variety of training arguments, such as optimizer and the regularization for $\triangle \boldsymbol{AB}$. Suppose that we use the AdamW (Loshchilov & Hutter, 2019) and $L_2$ regularization, the $\triangle \boldsymbol{AB}$ can be expanded in the following equation:

$$
\begin{aligned}
\triangle \boldsymbol{AB} &= \boldsymbol{A}^{(t+1)} \boldsymbol{B}^{(t+1)} - \boldsymbol{A}^{(t)} \boldsymbol{B}^{(t)} \\
&= (\boldsymbol{A}^{(t)} - \eta \widehat{g}_{\boldsymbol{A}} - \eta \beta \boldsymbol{A}^{(t)}) \cdot (\boldsymbol{B}^{(t)} - \eta \widehat{g}_{\boldsymbol{B}} - \eta \beta \boldsymbol{B}^{(t)}) - \boldsymbol{A}^{(t)} \boldsymbol{B}^{(t)} \\
&= \boldsymbol{A}^{(t)} \boldsymbol{B}^{(t)} - \eta \boldsymbol{A}^{(t)} \widehat{g}_{\boldsymbol{B}} - \eta \beta \boldsymbol{A}^{(t)} \boldsymbol{B}^{(t)} - \eta \widehat{g}_{\boldsymbol{A}} \boldsymbol{B}^{(t)} + \eta^2 \widehat{g}_{\boldsymbol{A}} \widehat{g}_{\boldsymbol{B}} + \eta^2 \beta \widehat{g}_{\boldsymbol{A}} \boldsymbol{B}^{(t)} \\
&\quad - \eta \beta \boldsymbol{A}^{(t)} \boldsymbol{B}^{(t)} + \eta^2 \beta \boldsymbol{A}^{(t)} \widehat{g}_{\boldsymbol{B}} + \eta^2 \beta^2 \boldsymbol{A}^{(t)} \boldsymbol{B}^{(t)} - \boldsymbol{A}^{(t)} \boldsymbol{B}^{(t)} \\
&= -\eta \boldsymbol{A}^{(t)} \widehat{g}_{\boldsymbol{B}} - \eta \beta \boldsymbol{A}^{(t)} \boldsymbol{B}^{(t)} - \eta \widehat{g}_{\boldsymbol{A}} \boldsymbol{B}^{(t)} + \eta^2 \widehat{g}_{\boldsymbol{A}} \widehat{g}_{\boldsymbol{B}} + \eta^2 \beta \widehat{g}_{\boldsymbol{A}} \boldsymbol{B}^{(t)} \\
&\quad - \eta \beta \boldsymbol{A}^{(t)} \boldsymbol{B}^{(t)} + \eta^2 \beta \boldsymbol{A}^{(t)} \widehat{g}_{\boldsymbol{B}} + \eta^2 \beta^2 \boldsymbol{A}^{(t)} \boldsymbol{B}^{(t)} \\
&\approx -\eta \boldsymbol{A}^{(t)} \widehat{g}_{\boldsymbol{B}} - \eta \widehat{g}_{\boldsymbol{A}} \boldsymbol{B}^{(t)}
\end{aligned}
\tag{8}
$$

where $\eta$ is the learning rate, $\beta$ is weight decay. What's more, for pre-trained weight $\boldsymbol{W}$, $\triangle \boldsymbol{W} = \eta \widehat{g}_{\boldsymbol{W}} + \eta \beta \boldsymbol{W}^{(t)}$. As a consequence, $\triangle \boldsymbol{AB}$ is not equal to $\triangle \boldsymbol{W}$ in the training process.

### A.5 CHANGE THE LEARNING RATE AND START STEPS TO SHOW BETTER PERFORMANCE.

We explored better hyper parameters of our Delta-LoRA by modifying the learning rate and trying more start steps and update ratio to prove the effectiveness of Delta-LoRA.

Table 8: The evaluation results of our proposed Delta-LoRA by using better hyper-parameters on GLUE benchmark.

| Method | MNLI | SST-2 | MRPC | CoLA | QNLI | QQP | RTE | STS-B | AVG |
|--------|------|-------|------|------|------|-----|-----|-------|-----|
| LoRA | 87.40 | 94.61 | 89.95 | 63.17 | 93.02 | 90.67 | 86.64 | 91.54 | 87.12 |
| DyLoRA | 86.33 | 94.26 | 89.46 | 61.12 | 92.22 | 90.17 | 84.47 | 91.06 | 86.14 |
| AdaLoRA ¶ | 87.34 | 94.49 | 90.19 | 61.64 | 93.08 | 90.14 | 85.19 | 91.16 | 86.65 |
| Delta-LoRA | 87.62±0.21 | 95.29±0.23 | 90.60±0.14 | 64.64±0.86 | 93.09±0.15 | 91.01±0.06 | 87.00±0.36 | 91.61±0.04 | 87.60 |

Table 9: The better training hyper-parameters that we obtained of our proposed Delta-LoRA on GLUE benchmark.

| Hyper-Parameter | MNLI | SST-2 | MRPC | CoLA | QNLI | QQP | RTE | STS-B |
|-----------------|------|-------|------|------|------|-----|-----|-------|
| Learning Rate $\eta$ | 4e-4 | 5e-4 | 5e-4 | 6e-4 | 3e-4 | 6e-4 | 4e-4 | 4e-4 |
| Start Steps $K$ | 2000 | 400 | 10 | 200 | 600 | 400 | 200 | 200 |
| Update Ratio $\lambda$ | 0.5 | 0.5 | 0.5 | 1 | 1 | 0.5 | 0.5 | 0.5 |

## A.6 THE PARAMETER SENSITIVITY STUDY

Table 10: The parameter sensitivity study of update ratio $\lambda$ for our proposed Delta-LoRA on E2E Challenge dataset. The best results are **boldfaced**.

| $\lambda$ | BLEU | NIST | METEOR | ROUGE-L | CIDEr |
|---|---|---|---|---|---|
| 0 | 68.94 | 8.73 | 45.27 | 70.81 | 2.41 |
| 1 | 69.77 | 8.81 | 45.99 | 71.58 | 2.46 |
| 2 | **70.84** | **8.91** | **46.47** | **72.24** | **2.53** |
| 3 | 70.14 | 8.84 | 46.39 | 71.45 | 2.45 |
| 4 | 70.03 | 8.83 | 46.21 | 71.56 | 2.47 |
| 5 | 70.13 | 8.85 | 46.35 | 71.72 | 2.48 |

**Parameter Sensitivity.** Here, we explore the hyper-parameter $K$ in Algorithm 1 and $\lambda$ in Equation 6. For the hyper-parameter $K$, we select it from 0 to 1000 with the interval of 100. From Table 11, we find that our Delta-LoRA could not bring in any improvement before $K = 400$, and it will keep a relatively good performance when $K$ is larger than 500. What is more, we choose different numbers for $\lambda$, ranging from 0 to 5. According to Table 10, the 5 metrics rise rapidly after $\lambda = 0$ and reach best at $\lambda = 2$, while the performance has small drops on 5 evaluation scores if $\lambda$ is chosen from 3 to 5.

Table 11: The parameter sensitivity study of start steps $K$ for our proposed Delta-LoRA on E2E Challenge dataset. The best results are **boldfaced**.

| $K$ | BLEU | NIST | METEOR | ROUGE-L | CIDEr |
|---|---|---|---|---|---|
| 0 | 69.10 | 8.75 | 45.54 | 71.31 | 2.41 |
| 100 | 69.97 | 8.84 | 46.07 | 71.40 | 2.46 |
| 200 | 69.72 | 8.83 | 45.82 | 71.41 | 2.43 |
| 300 | 69.73 | 8.86 | 45.98 | 71.09 | 2.46 |
| 400 | 70.18 | 8.89 | 46.30 | 71.66 | 2.49 |
| 500 | 70.84 | 8.91 | 46.47 | **72.24** | **2.53** |
| 600 | 70.38 | 8.86 | 46.38 | 71.70 | 2.47 |
| 700 | 70.61 | 8.89 | 46.43 | 72.13 | 2.51 |
| 800 | 70.70 | 8.89 | 46.30 | 71.97 | 2.51 |
| 900 | **71.00** | **8.92** | **46.47** | 72.04 | 2.52 |
| 1000 | 70.87 | 8.89 | 46.31 | 72.06 | 2.50 |

## A.7 HYPER-PARAMETER USED IN OUR EXPERIMENTS

We report the hyper-parameter that used in our experiments. Table 12 and Table 13 show the hyper-parameter that we used for the training and evaluation on E2E Challenge and WebNLG Challenge 2017 dataset. The Table 14 and Table 15 are the training and evaluation hyper parameter for XSum dataset, and the Table 16 consists of hyper-parameters for 8 datasets in GLUE benchmark.

Table 12: The training hyper-parameter used for E2E Challenge and WebNLG Challenge 2017 dataset.

| Hyper-Parameter | E2E Challenge | WebNLG Challenge 2017 |
|---|---|---|
| Learning Rate $\eta$ | 2e-4 | 2e-4 |
| Batch Size | 8 | 8 |
| Number of Epochs | 5 | 5 |
| Weight Decay $\beta$ | 0.01 | 0.01 |
| Resid_pdrop | 0 | 0.09 |
| Attn_pdrop | 0 | 0.09 |
| Embd_pdrop | 0 | 0 |
| Label Smooth | 0 | 0 |
| Start Steps $K$ | 500 | 500 |
| Update Ratio $\lambda$ | 2 | 5 |
| Rank $r$ | 4 | 4 |
| Alpha $\alpha$ | 32 | 32 |
| Trainable Matrices | $\boldsymbol{W}_Q, \boldsymbol{W}_V$ | $\boldsymbol{W}_Q, \boldsymbol{W}_V$ |
| LR Scheduler | Linear | Linear |
| Warmup Steps | 500 | 500 |

Table 13: The hyper-parameter for evaluation used for E2E Challenge and WebNLG Challenge 2017 dataset.

| Hyper-Parameter | E2E Challenge | WebNLG Challenge 2017 |
|---|---|---|
| Beam Size | 10 | 5 |
| Penalty | 0.8 | 1.0 |
| No Repeat Ngram Size | 4 | 4 |

Table 14: The training hyper-parameter used for XSum dataset.

| Hyper-Parameter | Xsum |
|---|---|
| Learning Rate $\eta$ | 2e-4 |
| Batch Size | 64 |
| Number of Epochs | 25 |
| Weight Decay $\beta$ | 0 |
| Activation Dropout | 0 |
| Dropout | 0 |
| Classifier Dropout | 0 |
| Start Steps $K$ | 1000 |
| Update Ratio $\lambda$ | 0.5 |
| Rank $r$ | 4 |
| Alpha $\alpha$ | 32 |
| Trainable Matrices | $\boldsymbol{W}_Q, \boldsymbol{W}_V$ |
| LR Scheduler | Linear |
| Warmup Steps | 3000 |

Table 15: The hyper-parameter for evaluation used for XSum dataset.

| Hyper-Parameter | Xsum |
|---|---|
| Beam Size | 8 |
| Penalty | 1.0 |
| No Repeat N-gram Size | 4 |

Table 16: The training hyper-parameters of our proposed Delta-LoRA on GLUE benchmark. We adopt the most of hyper-parameters in LoRA's paper and implement our method based on the codes given by LoRA's repository.

| Hyper-Parameter | MNLI | SST-2 | MRPC | CoLA | QNLI | QQP | RTE | STS-B |
|---|---|---|---|---|---|---|---|---|
| Learning Rate $\eta$ | 5e-4 | 5e-4 | 4e-4 | 4e-4 | 4e-4 | 4e-4 | 4e-4 | 4e-4 |
| Batch Size | 128 | 128 | 128 | 64 | 256 | 128 | 128 | 128 |
| Number of Epochs | 30 | 60 | 30 | 80 | 25 | 25 | 80 | 40 |
| Weight Decay $\beta$ | 0.1 | 0.1 | 0.1 | 0.1 | 0.1 | 0.1 | 0.1 | 0.1 |
| Max Sequence Length | 256 | 256 | 256 | 256 | 256 | 256 | 512 | 256 |
| Start Steps $K$ | 2000 | 400 | 10 | 100 | 800 | 400 | 200 | 200 |
| Update Ratio $\lambda$ | 0.5 | 0.5 | 0.5 | 0.5 | 0.5 | 0.5 | 0.5 | 0.5 |
| Rank $r$ | 8 | 8 | 8 | 8 | 8 | 8 | 8 | 8 |
| Alpha $\alpha$ | 16 | 16 | 16 | 16 | 16 | 16 | 16 | 16 |
| LR Scheduler | Linear | Linear | Linear | Linear | Linear | Linear | Linear | Linear |
| Trainable Matrices | $\boldsymbol{W}_Q,\boldsymbol{W}_V$ | $\boldsymbol{W}_Q,\boldsymbol{W}_V$ | $\boldsymbol{W}_Q,\boldsymbol{W}_V$ | $\boldsymbol{W}_Q,\boldsymbol{W}_V$ | $\boldsymbol{W}_Q,\boldsymbol{W}_V$ | $\boldsymbol{W}_Q,\boldsymbol{W}_V$ | $\boldsymbol{W}_Q,\boldsymbol{W}_V$ | $\boldsymbol{W}_Q,\boldsymbol{W}_V$ |
| Warmup Ratio | 0.06 | 0.06 | 0.06 | 0.06 | 0.06 | 0.06 | 0.06 | 0.06 |
| Evaluation Metrics | Accuracy | Accuracy | Accuracy | Matthews Correlation | Accuracy | Accuracy | Accuracy | Pearson |

