# OpenReview forum: "Delta-LoRA: Fine-Tuning High-Rank Parameters with the Delta of Low-Rank Matrices"
_ICLR.cc/2024/Conference — Submitted to ICLR 2024_

### Official Review · Reviewer_SSSB · 2023-10-26

**Soundness:** 3 good
**Presentation:** 3 good
**Contribution:** 2 fair
**Rating:** 6
**Confidence:** 3

**Summary:**

This paper introduces a new efficient fine-tuning method for large language models. Compared to LoRA, Delta-LoRA will fine-tune both the adapter and the original weight of the model.

**Strengths:**

The writing of this paper is good, making the paper easy to follow.
The method is reasonable and easy to reproduce, and the experiments demonstrate the efficiency of the method.
The challenge with directly fine-tuning a large language model is the memory cost. This method provides a way to update LoRA to fine-tune the backbone. In this case, the model can train the backbone directly without incurring a significant GPU memory cost.

**Weaknesses:**

Since LoRA is a framework, it would be beneficial if the author could apply Delta-LoRA to more large language models.
The paper doesn't clearly explain why Delta-LoRA works.
In terms of novelty and significance, I believe it is borderline.

**Questions:**

Can the author apply Delta-LoRA to llama2 and provide the experimental results?

---

> ### Author Response · Authors · 2023-11-21
> **Author Response to Reviewer SSSB**
>
> We are grateful for your constructive comments. We have revised our paper according to your constructive suggestions.
>
>
> **Strengths:**
> *The writing of this paper is good, making the paper easy to follow. The method is reasonable and easy to reproduce, and the experiments demonstrate the efficiency of the method. The challenge with directly fine-tuning a large language model is the memory cost. This method provides a way to update LoRA to fine-tune the backbone. In this case, the model can train the backbone directly without incurring a significant GPU memory cost.*
>
> **Question 1:** *Since LoRA is a framework, it would be beneficial if the author could apply Delta-LoRA to more large language models. The paper doesn't clearly explain why Delta-LoRA works. In terms of novelty and significance, I believe it is borderline.*
>
> **Our Reply:**
>
>
> We have used a variety of language models(e.g. GPT-2 and Bart-Large as well as LLaMA-7B with 7 Billions parameters) to conduct experiments with our Delta-LoRA. Based on the experimental results, we believe that our Delta-LoRA can generalize well on other LLMs.
>
>
> We would like to clarify the following three strengths of Delta-LoRA.
> - **Mathmaticlly, $g_W =  g_{AB}$.** Therefore, it is reasonable to update the pretrained $W$ by $\Delta AB$.
> - **Delta-LoRA showcases the lowest cosine similarity with the pretrained model.** According to our understanding experiment in Section 5.4, we found that Delta-LoRA has the lowest similarity with the pretrained model among all baselines, underscoring that our approach induces the most significant modifications to the final matrix $\boldsymbol{W}^{∗}$.
> - **Delta-LoRA shows better performance compared with LoRA on LLaMA-7B.** According to the evaluation experiment in our paper, we found that Delta-LoRA can yield higher evaluation result than LoRA with Alpaca dataset.
>
> **Question 2:** *Can the author apply Delta-LoRA to llama2 and provide the experimental results?*
>
> **Our Reply:**
> Thank you very much for your suggestion. We have tested the LLaMA-7B, following the setting in most papers. The results can be found in Section 5.1 in the revised manuscript. We can see that our Delta-LoRA has around 10\% improvement compared with LoRA evaluated by LLMs.
>
> *We will really appreciate if you can reconsider your score based on our revisions.*

---

> ### Author Response · Authors · 2023-11-23
> **Additional Questions?**
>
> *We hope our response addresses your concerns; if so, we would really appreciate it if you would reconsider your score accordingly. Please let us know if you have additional questions.*

---

> > ### Comment · Reviewer_SSSB · 2023-11-23
> >
> > Thank you for the author's response; I would like to maintain my score.

---

### Official Review · Reviewer_e3re · 2023-10-30

**Soundness:** 2 fair
**Presentation:** 3 good
**Contribution:** 2 fair
**Rating:** 5
**Confidence:** 3

**Summary:**

This paper aims to improve LoRA by also updating the pretrained weights $W$ in addition to the low-rank adapters $A$ and $B$: $W^{(t+1)} = W^{(t)} + \text{coefficient} \cdot (A^{(t+1) B^{(t+1)} - A^{(t) B^{(t)}})$. Experiments are conducted to compare the proposed method (Delta-LoRA) and other baselines including LoRA, DyLoRA, and AdaLoRA

**Strengths:**

1. The idea is very simple but seems to be effective as shown in the experiments.
2. The paper is easy to follow.

**Weaknesses:**

1. [Major] **Delta-LoRA seems to be fundamentally equivalent to LoRA.** I am very unclear about the fundamental difference between updating the pretrained weights using Delta-LoRA and LoRA. Basically, **both of them are performing low-rank updates**. Updating the pretrained weights does not seem to have any benefits, and if we treat W+AB as a whole $\hat{W}$, then, what you are doing can be viewed as LoRA but applied with a different learning rate. Please correct me if I am wrong.
2. [Major] **Many experiment details are not justified.**
    1. Neither DyLoRA nor AdaLoRA outperforms LoRA as they claimed in their paper. I noticed that the authors chose different settings with the AdaLoRA, this might be the reason for the bad performance. However, the authors do not provide any explanation on why they chose this different setting. This makes the outperformance of Delta-LoRA reported in the tables less convincing.
    2. Why does the author only try the experiments with cases with rank = 8? How about other cases?
    3. Why is the learning rate fixed? In my intuition, the only difference between the Delta-LoRA seems to be the resulting learning rate as I explained in the first bullet point. Therefore, it is unclear whether the reported performance of Delta-LoRA happen to outperform LoRA in this learning rate.
    4. Why are the trainable parameters of Delta-LoRA set to be higher than LoRA in multiple experiments? (Table 1, 4). It is very unclear whether the improvement comes from the additional trainable parameters or extra updatable parameters (or dropouts).
3. [Medium] The improvements look not significant.
4. [Medium] The code is not publically available thus I cannot verify the reproducibility of the experiment results.
5.  [Minor] Listing extra updatable parameters as a column in the tables looks very confusing. For example, Table 4 made me think that Delta-LoRA uses 72M updatable parameters but achieves worse results than Fine-Tuning in the third row.

**Questions:**

See the weakness section.

---

> ### Author Response · Authors · 2023-11-21
> **Author Response to Reviewer e3re (Part 1)**
>
> Thank you very much for your valuable and constructive comments.
>
> **Question:**
> *[Major] Delta-LoRA **seems to be fundamentally equivalent to LoRA**. I am very unclear about the fundamental difference between updating the pretrained weights using Delta-LoRA and LoRA. Basically, **both of them are performing low-rank updates**. Updating the pretrained weights does not seem to have any benefits, and if we treat W+AB as a whole, then, what you are doing can be viewed as LoRA but applied with a different learning rate. Please correct me if I am wrong.*
>
>
>
>
> **Our Reply:**
> We would like to clarify some points and help you understand our Delta-LoRA better. We want to point out that Delta-LoRA is fundamentally different from LoRA in the following three aspects:
>
>  - Given $\boldsymbol{W + AB}$, $\boldsymbol{W}$ is fixed in LoRA, while $\boldsymbol{W}$ will be updated in our Delta-LoRA. This is **the largest difference between LoRA and Delta-LoRA**. This modification can enhance the representative ability of the model.
>  - $\text{Rank}(\Delta \boldsymbol{W_{Delta-LoRA}}) = \text{Rank}(\boldsymbol{W}^{(T)} - \boldsymbol{W}^{(0)} + \boldsymbol{AB})>\text{Rank}(\Delta \boldsymbol{W}_{LoRA}) = \text{Rank}(\boldsymbol{AB})$. The rank of the learned incremental weight matrix in our Delta-LoRA is larger than that in the original LoRA.
>  - The gradient flow is different between LoRA and Delta-LoRA. We can know that $\frac{\partial \mathcal{L}}{\partial A} = B^{\top}\cdot\frac{\partial \mathcal{L}}{\partial W}$ and $\frac{\partial \mathcal{L}}{\partial B} = \frac{\partial \mathcal{L}}{\partial W}\cdot A^{\top}$ according to the chain rule.
>
> Here is the **back-propagation process of LoRA**: $\frac{\partial \mathcal{L}}{\partial A^{(t+1)}}= \frac{\partial \mathcal{L}}{\partial W^{(t+1)}}\cdot (B^{(t+1)})^{\top}=\frac{\partial \mathcal{L}}{\partial W^{(0)}+(A^{(t)}+\Delta A^{(t)})(B^{(t)} + \Delta B^{(t)})}\cdot (B^{(t)}+\Delta (B^{(t)}))^{\top}$
> $\frac{\partial \mathcal{L}}{\partial B^{(t+1)}} = (A^{(t+1)})^{\top} \cdot \frac{\partial \mathcal{L}}{\partial W^{(t+1)}}=(A^{(t)}+\Delta (A^{(t)}))^{\top} \cdot \frac{\partial \mathcal{L}}{\partial W^{(0)}+(A^{(t)}+\Delta A^{(t)})(B^{(t)} + \Delta B^{(t)})}$
> Here is the **back-propagation process of Delta-LoRA**:
> $\frac{\partial \mathcal{L}}{\partial A^{(t+1)}} = \frac{\partial \mathcal{L}}{\partial W^{(t+1)}}\cdot (B^{(t+1)})^{\top}=\frac{\partial \mathcal{L}}{\partial ((W^{(t)}+ \lambda \Delta A^{(t)}B^{(t)}) + (A^{(t)}+\Delta A^{(t)})(B^{(t)}+\Delta B^{(t)})) }\cdot (B^{(t)}+\Delta (B^{(t)}))^{\top}$
> $\frac{\partial \mathcal{L}}{\partial B^{(t+1)}} = (A^{(t+1)})^{\top} \cdot \frac{\partial \mathcal{L}}{\partial W^{(t+1)}} = (A^{(t)}+\Delta (A^{(t)}))^{\top} \cdot \frac{\partial \mathcal{L}}{\partial ((W^{(t)}+ \lambda\Delta A^{(t)}B^{(t)}) + (A^{(t)} + \Delta A^{(t)})(B^{(t)} + \Delta B^{(t)})  )}$.
>
>
>
>
> **Question 2.1:** *Neither DyLoRA nor AdaLoRA outperforms LoRA as they claimed in their paper. I noticed that the authors chose different settings with the AdaLoRA, this might be the reason for the bad performance. However, the authors do not provide any explanation on why they chose this different setting. This makes the outperformance of Delta-LoRA reported in the tables less convincing.*
>
>
>
> **Our Reply:**
>
> ***For a fair comparison, we use the same training setups as LoRA. The AdaLoRA and DyLoRA chose different training settings as LoRA. In the following, we want to clarify the differences among them.***
>
> First, we want to point out that
>
> 1. DyLoRA aims to improve the performance of lower rank when inference rather than the rank they selected at the training stage. That means the performance they claimed in their paper was based on a higher rank, which can yield a much more memory overhead. For a fair comparison, we report their results based on the low-rank setting.
>
> Second, AdaLoRA uses a different setting from LoRA, the differences are summarized as:
>
> 1. The AdaLoRA fine-tuned each layer of the pretrained matrices in Transformer. This operation will cause extra **activation memory** consumption (i.e. **19.79 GB extra GPU memory** for LLaMA-7B with batch size of 1024 and 1024 max length implemented by Pytorch). Therefore, we do not choose the similar training setups with AdaLoRA. Instead, we used the same training setups with LoRA.
> 2. AdaLoRA has 7 extra hyper parameters (i.e. target_rank, reg_orth_coef, init_warmup, final_warmup, mask_interval, beta1, beta2). We cannot search the best hyper-parameters for the different datasets and models.

---

> ### Author Response · Authors · 2023-11-21
> **Author Response to Reviewer e3re (Part 2)**
>
> **Question 2.2:** *Why does the author only try the experiments with cases with rank = 8? How about other cases?*
>
> **Our Reply:**
>
> Thanks for your constructive comments. We would like to emphasize that we use the same training setups with LoRA. We use the rank=8 on 8 datasets in GLUE benchmark, while rank=4/8 for NLG tasks. *We explore more ranks in the following works. The table in the following is the experiment that we use different ranks on CoLA dataset.*
>
> | Rank | LoRA | Delta-LoRA |
> |----------|----------|----------|
> | 8 | 63.17 | **63.82** |
> | 16 | 63.06 | **64.30** |
> | 32 | 63.81 | **64.79** |
> | 64 | 63.39 | **65.59** |
>
> We can see a better choice of rank can further improve Delta-LoRA.
>
> **Question 2.3:** *Why is the learning rate fixed? In my intuition, the only difference between the Delta-LoRA seems to be the resulting learning rate as I explained in the first bullet point. Therefore, it is unclear whether the reported performance of Delta-LoRA happen to outperform LoRA in this learning rate.*
>
> **Our Reply:**
>
> To make fair comparison, we use the same learning rate as reported in the original paper and code(e.g. 2e-4 on E2E Challenge dataset). Besides, we apply linear learning rate scheduler at the training stage, which is same as the setting of LoRA. We could use better hyper-parameters to further improve the results of our Delta-LoRA with standard errors.
>
>
> | MNLI| SST-2 | MRPC | CoLA | QNLI | QQP | RTE | STS-B | AVG|
> |----------|----------|----------|----------|----------|----------|----------|----------|----------|
> | 87.62 $\pm$ 0.21 | 95.29 $\pm$ 0.23 | 90.60 $\pm$ 0.14 | 64.64 $\pm$ 0.86 | 93.09 $\pm$ 0.15 | 91.01 $\pm$ 0.06 | 87.00 $\pm$ 0.36 | 91.61 $\pm$ 0.04 | 87.60 |
>
>
>
>
> **Question 2.4:** *Why are the trainable parameters of Delta-LoRA set to be higher than LoRA in multiple experiments? (Table 1, 4). It is very unclear whether the improvement comes from the additional trainable parameters or extra updatable parameters (or dropouts).*
>
> **Our Reply:**
>
> The **Extra Updatable Trainable Parameters** means the adjustable pretrained parameters, which we update by using the delta of low rank matrices, without extra cost of GPU resources. For example, since LoRA couldn't update the pretrained $\boldsymbol{W}$, so that its **Extra Updatable Trainable Parameters** are **0**. The Table 6 shows the **Delta-LoRA+LoRA module**, which means that we add the Dropout layer in Delta-LoRA module. As we reported, only using the Delta-LoRA update strategy indeed prompts the model to have better performance, while removing the Dropout layer can have more performance gain.
>
>
> **Question 3.1:** *[Medium] The improvements look not significant.*
>
> **Our Reply:**
>
> We have decent improvement on most NLG datasets (e.g. 1.24 on E2E, 0.91 on WebNLG, 10.4\% on LLaMA-7B with Alpaca dataset), and some NLU datasets (e.g. 0.65 on CoLA). More importantly, we use the same learning rate and same model architecture with LoRA, we do not search the best hyper-parameter and show the real performance of our Delta-LoRA in our paper. The Delta-LoRA has consistent performance improvement over 4 NLG datasets and 8 NLG datasets, which is a general method and can be applied to multiple domains.
>
>
> **Question 3.2:** *[Medium] The code is not publically available thus I cannot verify the reproducibility of the experiment results.*
>
>
> **Our Reply:**
>
> We will release our models and all codes after the decision of this conference.
>
>
>
> **Question 3.3:** *[Minor] Listing extra updatable parameters as a column in the tables looks very confusing. For example, Table 4 made me think that Delta-LoRA uses 72M updatable parameters but achieves worse results than Fine-Tuning in the third row.*
>
>
> **Our Reply:**
>
> We would like to modify our paper to clarify the main ideas and contributions as well. Generally, **The Extra Updatable Trainable Parameters** claimed in our paper means the adjustable pretrained parameters **but does not consume more memory**, which we update by using the delta of low rank matrices.
>
> *We will really appreciate if you can reconsider your score based on our revisions.*

---

> > ### Comment · Reviewer_e3re · 2023-11-22
> >
> > Q1: Thanks for your clarification! Now I understand that Delta-LoRA is not fundamentally equivalent to LoRA.
> >
> > Q2.1: Thanks for the clarification. Delta-LoRA uses less memory overhead than DyLoRA. Then DyLoRA is probably not a good baseline? Then I think the authors should use other common parameter-efficient fine-tuning methods such as (IA)^3 proposed by “[Few-Shot Parameter-Efficient Fine-Tuning is Better and Cheaper than In-Context Learning](https://arxiv.org/abs/2205.05638)”.
> >
> > Also, directly using LoRA’s hyperparameters for AdaLoRA looks a bit unfair since the hyperparameters might be optimized for LoRA, not the other methods. Therefore, this comparison is not that convincing. I feel like more experiments are needed for giving a more convincing conclusion.
> >
> > Q2.2: Thanks for running experiments on COLA. It seems on full-rank, Delta-LoRA outperforms the full fine-tuning (64.57). Do you have any intuition about why this happens? Otherwise, I think more experiments are important to empirically understand everything happening here.
> >
> > Q2.3: Thanks for the additional experiments. It seems hyperparameters still have a fair impact on the performance.
> >
> > Q2.4: How about Table 4?
> >
> > In summary, my major concerns still remain, and I will keep my original score for now.

---

> > > ### Author Response · Authors · 2023-11-22
> > > **Author Response to Reviewer e3re (Part 1)**
> > >
> > > We would like to thank you for your reply to our revised paper. We provide more analyses and comparisons for you. Hope you will be satisfied with our response.
> > >
> > >
> > > *Q1: Thanks for your clarification! Now I understand that Delta-LoRA is not fundamentally equivalent to LoRA.*
> > >
> > >
> > > **Our Reply:**
> > > Thanks for your reply! We are really happy that you are satisfied with our clarification of the novelty of Delta-LoRA.
> > >
> > >
> > >
> > > *Q2.1: Thanks for the clarification. Delta-LoRA uses less memory overhead than DyLoRA. Then DyLoRA is probably not a good baseline? Then I think the authors should use other common parameter-efficient fine-tuning methods such as (IA)^3 proposed by “Few-Shot Parameter-Efficient Fine-Tuning is Better and Cheaper than In-Context Learning”.*
> > >
> > > **Our Reply:**
> > > Thanks a lot for your further comments.
> > > To answer your question, we would like to point out the following two reasons:
> > >
> > > - Since the LoRA is one of the most powerful and popular PEFT methods according to the evaluation of Ding et. al. on 100 NLP tasks. LoRA is a good baseline, therefore, we follow the settings of LoRA. We compare Delta-LoRA with AdaLoRA and DyLoRA because both they change the ranks dynamically.
> > >
> > > - $(IA)^3$ learns to scale activations by some learned vectors. In our paper, we want to induce more parameters into the optimization process. The two methods are orthotropic and complimentary.
> > >
> > >
> > > Besides, we introduce more PEFT methods and compare with our Delta-LoRA. The experiments are conducted on WebNLG 2017 Challenge. $^*$ means the results are directly taken from the original paper (Prefix-Tuning, Li et al.).
> > >
> > > |Method | | BLEU&uarr;| | | MET&uarr;| | | TER&darr;| |
> > > |------------------------|------------------------|------------------------|------------------------|------------------------|------------------------|------------------------|------------------------|------------------------|------------------------|
> > > ||Seen | Unseen| All| Seen| Unseen| All| Seen| Unseen| All|
> > > |FT-Top2$^*$| 53.6| 18.9| 36.0| 0.38| 0.23| 0.31| 0.49| 0.99| 0.72|
> > > |Adapter-0.1\%$^*$ |54.5| 45.1| 50.2| 0.39| 0.36| 0.38| 0.40| 0.46| 0.43|
> > > |Prefix-Tuning$^*$|**62.9**| 45.6| 55.1| 0.44| 0.38| 0.41| 0.35| 0.49| 0.41|
> > > |Delta-LoRA| 62.87| **47.68**| **55.96**| **0.45**| **0.39**| **0.42**| **0.34**| **0.48**| **0.40**|
> > >
> > > Hope you will be satisfied with our reply.

---

> > > ### Author Response · Authors · 2023-11-22
> > > **Author Response to Reviewer e3re (Part 2)**
> > >
> > > *Also, directly using LoRA’s hyperparameters for AdaLoRA looks a bit unfair since the hyperparameters might be optimized for LoRA, not the other methods. Therefore, this comparison is not that convincing. I feel like more experiments are needed for giving a more convincing conclusion.*
> > >
> > > **Our Reply:**
> > > Thank you for your comments. We would like to clarify more details to answer your concern:
> > >
> > > - LoRA kept similar training setups for different tasks on 8 NLU datasets, such as 4e-4 for both STSB and CoLA, while AdaLoRA searched more fine-grained learning rates, such as 2.2e-3 for STSB and 5e-4 for CoLA. It is very difficult to search the hyper-parameters for different methods on different datasets. We believe it would be better to keep an unified setting. Therefore, we use the setting used in LoRA. We believe it is fair.
> > >
> > > - LoRA and AdaLoRA use different models to conduct experiments. To fairly compare them, we used the same model architecture and the same training setups.
> > >
> > > - We searched more hyperparameters for AdaLoRA on the same datasest. $^*$ means using the provided learning rate by AdaLoRA. The results are shown in the following table:
> > >
> > > |Dataset| Learning Rate | Evaluation Results |  Init Warmups | Final Warmups|
> > > |---------------|---------------|----------------|---------------|----------------|
> > > |CoLA| 5e-4    | 60.12    |    1000    |   3000   |
> > > |CoLA| 5e-4    | 61.13    |   1000 | 5000 |
> > > |CoLA| 5e-4    | 60.31    |   2000 | 3000 |
> > > |CoLA| 5e-4    | 60.07    |   2000   |   5000   |
> > > |CoLA$^*$| 4e-4    | 61.07    |   1000   |   3000   |
> > > |CoLA$^*$| 4e-4    | 59.30    |   1000   |   5000   |
> > > |CoLA$^*$| 4e-4    | 59.81    |   2000   |   3000   |
> > > |CoLA$^*$| 4e-4    | 59.06    |   2000   |   5000   |
> > >
> > > |Dataset| Learning Rate | Evaluation Results |  Init Warmups | Final Warmups|
> > > |---------------|--------------|----------------|---------------|---------------|
> > > |STSB| 4e-4 | 91.06 | 100 | 1000|
> > > |STSB| 4e-4 | 91.06 | 100 | 300 |
> > > |STSB| 4e-4 | 91.05 | 100 | 500 |
> > > |STSB| 4e-4 | 91.07 | 200 | 1000|
> > > |STSB| 4e-4 | 91.05 | 200 | 300 |
> > > |STSB| 4e-4 | 91.09 | 200 | 500 |
> > > |STSB| 4e-4 | 91.17 | 300 | 1000|
> > > |STSB| 4e-4 | 91.22 | 300 | 500 |
> > > |STSB| 4e-4 | 91.12 | 500 | 500 |
> > > |STSB| 4e-4 | 91.23 | 500 | 1000|
> > > |STSB$^*$| 2.2e-3    |  91.22   | 100 | 500 |
> > > |STSB$^*$| 2.2e-3    |  91.30   | 100 | 300 |
> > > |STSB$^*$| 2.2e-3    |  91.24   | 100 | 1000|
> > > |STSB$^*$| 2.2e-3    |  91.10   | 200 | 500 |
> > > |STSB$^*$| 2.2e-3    |  91.43   | 200 | 300 |
> > > |STSB$^*$| 2.2e-3    |  91.32   | 200 | 500 |
> > > |STSB$^*$| 2.2e-3    |  91.25   | 500 | 500 |
> > > |STSB$^*$| 2.2e-3    |  91.31   | 500 | 1000|
> > > |STSB$^*$| 2.2e-3    |  91.40   | 300 | 500 |
> > > |STSB$^*$| 2.2e-3    |  91.04   | 300 | 300 |
> > > |STSB$^*$| 2.2e-3    |  91.20   | 300 | 1000|
> > >
> > > Note that the performance of the best hyperparameter for CoLA and STSB of Delta-LoRA are **64.64 $\pm$ 0.86** and **91.61 $\pm$ 0.06**, respectively.
> > >
> > > Hope you will be happy with our reply.
> > >
> > > *Q2.2: Thanks for running experiments on COLA. It seems on full-rank, Delta-LoRA outperforms the full fine-tuning (64.57). Do you have any intuition about why this happens? Otherwise, I think more experiments are important to empirically understand everything happening here.*
> > >
> > > **Our Reply:**
> > > Thank you for your comment. We believe the reasons may be due to the following aspects:
> > >
> > > - **The data gap between the CoLA data and the pretrained data is large.** If the training data in downstream task is much different from the pretrained dataset, the full-finetuned can defeat LoRA and other PEFT methods. However, if training data is similar to the pretrained dataset, the fully finetuned method can overfitted fast and suffer from catastrophic forgetting.
> > >
> > > - **The data scale of the CoLA data is relatively small.** If the data scale of the finetuned data is small, full finetuning mode will overfit the finetuned data quickly. In this way, Delta-LoRA and LoRA will show better results. This is consistent with our results.
> > >
> > > *Q2.3: Thanks for the additional experiments. It seems hyperparameters still have a fair impact on the performance.*
> > >
> > > **Our Reply:**
> > > Thanks for your comment! Beter hyper-parameter does bring better performance. Therefore, we reported both original hyperparameters and modified hyperparameters in our paper to show the real improvement of our Delta-LoRA.
> > >
> > > *Q2.4: How about Table 4?*
> > >
> > > **Our Reply:**
> > > Thanks for your comment. The XSum dataset used for Table 4 is non-sensitive for most hyperparameters (we found that start steps=1000 and 1400 have similar results for our Delta-LoRA), but it is still sensitive for learning rate. We found that LoRA achieves best when learning rate is 2e-4 among all baselines. Our implementation is based on the code provided by the AdaLoRA, which guarantees the fair comparison. In our paper, we reported the performance of AdaLoRA by using the original hyper-parameters given by AdaLoRA, and changed the learning rate to 2e-4 for LoRA and Delta-LoRA.
> > >
> > > **We hope you can reconsider your score based on our new reply.**

---

> > > > ### Comment · Reviewer_e3re · 2023-11-22
> > > > **Thanks for your response**
> > > >
> > > > Q2.4: I am confused — I mean in Table 4, LoRA and Delta-LoRA are using different trainable parameters, right?
> > > >
> > > > Based on the response, I am happy to increase my score to 5.

---

> > > > > ### Author Response · Authors · 2023-11-23
> > > > > **Autor Response to Reviewer e3re**
> > > > >
> > > > > *Q2.4: I am confused — I mean in Table 4, LoRA and Delta-LoRA are using different trainable parameters, right?*
> > > > >
> > > > >
> > > > > **Our Reply:**
> > > > > Thank you very much for your carefulness. Yes, you are right, the parameters trainable in LoRA and Delta-LoRA are exactly the same. It is a typo in the previous version, we have updated the manuscript correspondingly. Thanks again for your careful reading of our submission.
> > > > >
> > > > >
> > > > >
> > > > >
> > > > > *Based on the response, I am happy to increase my score to 5.*
> > > > >
> > > > > **Our Reply:**
> > > > > Thanks for your kindness. We are happy that we successfully solved your concerns with additional experiments. Please let us know if you have additional questions or concerns. Hope you will reconsider your score again.

---

### Official Review · Reviewer_rQDs · 2023-11-02

**Soundness:** 3 good
**Presentation:** 3 good
**Contribution:** 3 good
**Rating:** 6
**Confidence:** 4

**Summary:**

The paper proposes DeltaLORA, a novel variant of the popular PEFT technique LoRA, which updates not only the low rank adapter matrices, but also propagates updates to the underlying pretrained matrix $W$. Their strategy helps alleviate the problem of the being limited to low rank representations which may be insufficient to adapt to downstream tasks while maintaining computational parity with LoRA.

**Strengths:**

- The paper is well-organized and easy to follow.
- Extensive evaluation on NLU, NLG tasks as well as ablation studies to emphasize the efficacy of their formulation.
- The proposed algorithm is well-motivated, it builds very naturally upon the baseline LORA and differentiates itself against the baseline (Section 5.3) to answer a natural question - "Does the performance stem from the additional parameters or the updates applied to the underlying weight matrix?"
- It accomplishes it's goals without sacrificing the key advantages of LORA - low memory consumption

**Weaknesses:**

## Major

- The only major drawback seems to be that the differences in empirical results do not seem statistically significant (and do not have any errors provided). Given the incremental increase, it is not clear whether the performance of the algorithm stems from real gains or is noise from insufficient tuning.
## Minor

- The whole idea is pretty much a heuristic which works very well in practice (but the same can be said of LoRA)
- The overt emphasis on the fact that $g_W = g_{AB}$ seems unnecessary (specifically Section 4.1) since the results stems from fairly straightforward linear algebra and calculus. It might be better to reduce that section, and allocate space to more impactful discussion/experiments.

It is certainly true that the work's novelty is primarily in engineering a practical method, rather than introducing a new direction of research. The advantage of method is clearly empirical in nature.

I am very divided on accepting this paper simply due to the fact that all algorithms seem very close to each other in empirical results, the differences being marginal in most cases. However, due to the fact that this method does provide positive gains across the board, I vote to accept
the paper.

**Questions:**

- Can you provide standard errors on some of your empirical results?
- It would help to incorporate some experiments in Section 7 of the LoRA paper to help understand the kind of updates induced by DeltaLoRA

---

> ### Author Response · Authors · 2023-11-21
> **Author Response to Reviewer rQDs**
>
> Thanks for your detailed reviews and valuable advice for our submitted paper.
>
> **About the Strengths:**
>
> - *The paper is well-organized and easy to follow.*
>
> - *Extensive evaluation on NLU, NLG tasks as well as ablation studies to emphasize the efficacy of their formulation.*
>
> - *The proposed algorithm is well-motivated, it builds very naturally upon the baseline LORA and differentiates itself against the baseline (Section 5.3) to answer a natural question - "Does the performance stem from the additional parameters or the updates applied to the underlying weight matrix?"*
>
> - *It accomplishes its goals without sacrificing the key advantages of LORA - low memory consumption.*
>
>
> **Our Reply:**
>
> Thanks for your recognition, we are encouraged by your comments. We have made some corresponding revisions according to your comments and concerns.
>
> **Question 1:** *The only major drawback seems to be that the differences in empirical results do not seem statistically significant (and do not have any errors provided). Given the incremental increase, it is not clear whether the performance of the algorithm stems from real gains or is noise from insufficient tuning.*
>
>
>
> **Our Reply:**
> We would like to emphasize the effectiveness of the proposed Delta-LoRA from the following points:
>
>
> - **We kept the similar training settings as LoRA**, such as learning rate, weight decay, rank number, training epochs and random seed. We do not use a parameter search and select the random seed (we kept all experiments with random seed=0). Therefore, the improvement that claimed in our paper is convincing.
>
> - **We achieved significant improvement compared with AdaLoRA and DyLoRA.** We used the same training setting to evaluate the two methods. We observed that our method could have pronounced performance gain.
>
> - **We witnessed consistent improvement across multiple datasets.** According to our experiment, we achieved improvement across 8 NLU datasets and 4 NLG datasets, which proves that our method works well across different tasks.
>
> **Question 2:** *The whole idea is pretty much a heuristic which works very well in practice (but the same can be said of LoRA)*
>
> **Our Reply:**
> Thanks for your comment. Our initial motivation was indeed heuristic, while we gave a mathematical formulation and presented our analysis of its rationality. We supplied a more theoretical proof to further demonstrate the difference between Delta-LoRA and LoRA in Sec. A.4 of the revised version.
>
>
>
>
> **Question 3:** *The overt emphasis on the fact that
>  seems unnecessary (specifically Section 4.1) since the results stem from fairly straightforward linear algebra and calculus. It might be better to reduce that section, and allocate space to more impactful discussions/experiments. It is certainly true that the work's novelty is primarily in engineering a practical method, rather than introducing a new direction of research. The advantage of method is clearly empirical in nature.*
>
> **Our Reply:**
>
> Thanks for your kind advice. We appropriately reduce this statement in our revised paper. We modified some content in our revised paper, and marked these descriptions with blue color.
>
>
> **Question 4:** *Can you provide standard errors on some of your empirical results?*
>
> **Our Reply:**
>
> We provide the standard errors on GLUE benchmark in the following table. We used better learning rate and start steps here which can be found in the Section A.5 of our revised paper.
>
> | MNLI| SST-2 | MRPC | CoLA | QNLI | QQP | RTE | STS-B | AVG|
> |----------|----------|----------|----------|----------|----------|----------|----------|----------|
> | 87.62 $\pm$ 0.21 | 95.29 $\pm$ 0.23 | 90.60 $\pm$ 0.14 | 64.64 $\pm$ 0.86 | 93.09 $\pm$ 0.15 | 91.01 $\pm$ 0.06 | 87.00 $\pm$ 0.36 | 91.61 $\pm$ 0.04 | 87.60|
>
> A better choice of learning rate and start steps can further improve the performance of Delta-LoRA.
>
> **Question 5:** *It would help to incorporate some experiments in Section 7 of the LoRA paper to help understand the kind of updates induced by Delta-LoRA.*
>
> Thanks for your reminder. We moved these experiment tables to a more conspicuous place. We provided the ablation study and parameter sensitivity in Section 5.3 and Appendix A.5 with 4 experiments in the revised paper.
>
> *We will really appreciate if you can reconsider your score based on our revisions.*

---

> > ### Author Response · Authors · 2023-11-23
> > **Additional Questions?**
> >
> > *We hope our response addresses your concerns; if so, we would really appreciate it if you would reconsider your score accordingly. Please let us know if you have additional questions.*

---

### Official Review · Reviewer_p8Lv · 2023-11-02

**Soundness:** 2 fair
**Presentation:** 3 good
**Contribution:** 2 fair
**Rating:** 3
**Confidence:** 5

**Summary:**

This paper introduces Delta-LoRA. Unlike other low-rank adaptation techniques, it updates both low-rank matrices and pre-trained weights. The authors claim this approach tackles the issue of inadequate learning representations and matches LoRA in terms of memory and computational costs.

**Strengths:**

The paper is clear and easy to read with straightforward derivations. The proposed method, Delta-LoRA, is original in its approach to update both the low-rank matrices and the pre-trained weights, which could potentially address the limitations of existing low-rank adaptation methods.

**Weaknesses:**

The main weakness of the paper is the lack of experiments on large models where the memory efficiency of Delta-LoRA would be most beneficial. The models tested in the paper are relatively small, which makes the memory-saving advantage less meaningful. Additionally, there is a performance loss compared to fine-tuning, which is expected. However, the paper should have reported the performance of full finetuning with stateless optimizers (such as SGD) and showed that Delta-LoRA outperforms this baseline.

**Questions:**

- Could you provide more evidence to support the claim that Delta-LoRA can nearly match the performance of full finetuning with "large" models?
- How does Delta-LoRA's performance compare to full finetuning with stateless optimizers? (Note -- the memory footprint is actually slightly lower with full finetuning with stateless optimizers)

---

> ### Author Response · Authors · 2023-11-21
> **Author Response to Reviewer p8Lv**
>
> Thank you very much for recognizing the novelty of this paper. We have revised our paper according to your comments.
>
> **Strengths:** *The paper is clear and easy to read with straightforward derivations. The proposed method, Delta-LoRA, is original in its approach to update both the low-rank matrices and the pre-trained weights, which could potentially address the limitations of existing low-rank adaptation methods.*
>
> **Our Reply:**
> Delta-LoRA introduces a simple but efficient strategy to finetune the low-rank matrices and the pre-trained weights. Delta-LoRA improves the original LoRA from two perspectives: performance and memory consumption.
>
> **Question 1.1:** *The main weakness of the paper is the lack of experiments on large models where the memory efficiency of Delta-LoRA would be most beneficial. The models tested in the paper are relatively small, which makes the memory-saving advantage less meaningful.*
>
> **Our Reply:**
> Thanks a lot for your suggestions! We have evaluated the performance of LLaMA-7B model on Alpaca dataset, in the revised manuscript, we have clarified the experimental results. We find that, using evaluation of GPT-4, Delta-LoRA improves  LoRA by around 10\%, which proves the effectiveness of our Delta-LoRA.
>
>
>
>
>
> **Question 1.2:** *Additionally, there is a performance loss compared to fine-tuning, which is expected.*
>
> **Our Reply:**
> For this question, we would like to point out there exist two exceptions:
>
> 1. **The gap between the finetuned data and the pretrained data is large.** If the training data in downstream task is much different from the pretrained dataset, the full-finetuned can defeat LoRA and other PEFT methods. However, if training data is similar to the pretrained dataset, the fully finetuned method can overfitted fast and suffer from catastrophic forgetting.
> 2. **The data scale of the finetuned data is small**. If the data scale of the finetuned data is small, full finetuning mode will overfit the finetuned data quickly. In this way, Delta-LoRA and LoRA will show better results. This is consistent with our results.
>
>
>
> **Question 1.3 and Question 2.2:**
> - *The paper should have reported the performance of full finetuning with stateless optimizers (such as SGD) and showed that Delta-LoRA outperforms this baseline.*
> - *How does Delta-LoRA's performance compare to full finetuning with stateless optimizers? (Note -- the memory footprint is actually slightly lower with full finetuning with stateless optimizers)*
>
> **Our Reply:**
>
> The AdamW is widely used in the training of transformers, but SGD is not so popular. In NLP and multimodal tasks, most people choose AdamW instead of SGD, since it has a stable performance and converges fast. As the reasons described before, we chose the AdamW optimizer to conduct experiments to make fair comparison. We run the experiments with SGD optimizer on CoLA dataset with rank=8, $\alpha$=16, learning rate=0.01 and momentum=0.9. LoRA achieves 63.67 while Delta-LoRA has 64.32. Besides, we observed that the training process occurred crash after 57 epochs for the LoRA's training, which emphasizes the instablity of the SGD optimizer for current NLP tasks.
>
>
> **Question 2.1:** *Could you provide more evidence to support the claim that Delta-LoRA can nearly match the performance of full finetuning with "large" models?*
>
> **Our Reply:**
> We have tested the effectiveness of Delta-LoRA with LLaMA-7B on Alpaca dataset. We found that the memory cost of Delta-LoRA is basically same as the LoRA, while our method outperforms the LoRA for around 10\% under the evaluation of LLMs.
>
> *We will really appreciate if you can reconsider your score based on our revisions.*

---

> ### Author Response · Authors · 2023-11-23
> **Additional questions?**
>
> *We hope our response addresses your concerns; if so, we would really appreciate it if you would reconsider your score accordingly. Please let us know if you have additional questions.*

---

### Author Response · Authors · 2023-11-21
**Summary of Paper Revision**

We would like to thank all reviewers for their valuable and constructive comments.

We humbly accepted the reviewers's suggestions and modified our paper to further address reviewers' concerns. We believe the manuscript has been largely improved.

Specifically, we have made the following changes:

- We added more experiments to further prove the effectiveness of the proposed Delta-LoRA in Section A.5.

- We clarified our theoretical and empirical contributions more clearly in Sec. 5.1 and 5.3.

- We supplied a more theoretical proof to demonstrate the difference between Delta-LoRA and LoRA in Sec. A.4.

We mark the revised content in blue color for your reference.

---

### Meta-Review · Area_Chair_NG1t · 2023-12-16

**Metareview:**

The paper proposes a method called Delta-LoRA in which a pre-trained weight matrix W is updated with low-rank factors A and B. While the main idea resembles LoRA (and its derivates), the authors argue that by updating W with the difference of low-rank factors at successive iterations propagates learning to W as well.

Strengths:
+ The paper is well written and easy to read
+ Evaluations on different datasets
+ The idea is simple

Weaknesses
- Novelty of the paper is limited. It is difficult to argue the paper has a significant difference from LoRA (equivalent to LoRA if we consider the weights are updated with the low-rank factors and initialize the low-rank factors with zero at next iteration).
- Significance: Performance improvement offered by this method is small.

**Justification For Why Not Higher Score:**

- Delta-LoRA seems to be fundamentally equivalent to LoRA.
- Empirical results do not show significant improvement over existing LoRA-based methods

**Justification For Why Not Lower Score:**

N/A

---

### Decision · Program_Chairs · 2024-01-16

Reject